# A data-driven biology-based network model reproduces *C. elegans* premotor neural dynamics

Megan Morrison[1]*, Lai-Sang Young[2]

**1** Applied Mathematics, Illinois Institute of Technology, Chicago, Illinois, United States of America,
**2** Courant Institute of Mathematical Sciences, New York University, New York, New York, United States of America

* mmorrison3@illinoistech.edu

## Abstract

*C. elegans* locomotion is composed of switches between forward and reversal states punctuated by turns. This locomotory capability is necessary for the nematode to move towards attractive stimuli, escape noxious chemicals, and explore its environment. Although experimentalists have identified a number of premotor neurons as drivers of forward and reverse motion, how these neurons work together to produce the behaviors observed remains to be understood. Towards a better understanding of *C. elegans* neurodynamics, we present in this paper a minimally parameterized, biology-based dynamical systems model of the premotor network. Our model consists of a recurrently connected collection of premotor neurons (the core group) driven by over a hundred sensory and interneurons that provide diverse feedforward inputs to the core group. It is data-driven in the sense that the choice of neurons in the core group follows experimental guidance, anatomical structures are dictated by the connectome, and physiological parameters are deduced from whole-brain imaging and voltage clamps data. When simulated with realistic input signals, our model produces premotor activity that closely resembles experimental data: from the seemingly random switching between forward and reversal behaviors to the synchronization of subnetworks to various higher-order statistics. We posit that different roles are played by gap junctions and synaptic connections in switching dynamics. Using the model we identify signal neurons that strongly influence switches between behavioral states and core neurons that play an important role in integrating signal information. The model produces switching statistics that underlie behaviors such as dwelling versus roaming as a result of the synaptic inputs received.

**Data availability statement:** There are no primary data in the paper. The model weights and source code to reproduce the simulations and statistical analysis is available on GitHub at https://github.com/mmtree/Celegans_premotor.

**Funding:** This work was supported by the National Science Foundation, https://www.nsf.gov, (MM, award no. 2103239 and LSY, grants DMS-1901009 and DMS-2350184). The funders had no role in study design, data collection and analysis, decision to publish, or preparation of the manuscript.

**Competing interests:** The authors have declared that no competing interests exist.

## Author summary

The nematode *C. elegans* has one of the most complete connectomes and relatively few behavioral states, making it an ideal organism for which to attempt the formidable task of connecting biology to function. This paper is about the premotor network, which controls locomotion. We model premotor activity as a driven dynamical system: a recurrent network of ~15 neurons known to be intimately connected to forward and reversal movements, driven by over 100 of their presynaptic neurons. Whole-brain imaging data are used to constrain parameters. Comparing model outputs to data, we show that the model accurately reproduces basic features of *C. elegans* movements, such as the seemingly random switches between forward crawling and reversals. Because our model of neural activity is semi-realistic and analyzable, it has the potential to reveal mechanisms and predict behavior. As an example, we use it to clarify the role of gap junctions versus synaptic inputs and the dependence on various sensory neurons.

## Introduction

To survive, animals must be able to generate a wide range of behaviors and to switch between them effectively. These behaviors, ubiquitous across species, include searching for food, forming social aggregates, and escaping from danger. The aim of this paper is to study the neural mechanisms responsible for producing different behaviors and inducing transitions. *C. elegans*, with their relatively simple nervous system, limited behavioral states, and amenability to experimentation [1–3], are an ideal candidate to attempt a theory that connects neurobiology to behavior. The building of such a theory is an overarching goal of the present work.

The *C. elegans* connectome consists of 300+ neurons; interneurons act as the main processors of information, they receive input from sensory neurons and send commands to motor neurons [4–6]. To some extent, experimentalists have demystified how sensory neurons respond to stimuli and how motor neurons coordinate muscle activity [7–11], but relatively little is known about how the intermediary interneurons operate collectively to make behavioral decisions. A subset of interneurons—premotor neurons—are chiefly responsible for determining the most common locomotory behaviors [5,6,9]. In this work, we model and analyze the activity of the premotor network of *C. elegans*.

Despite the remarkable progress that has been made in documenting various aspects of *C. elegans* neural circuits, gaps in our knowledge remain. Individual neurons that promote forward and reversal movements have been identified, but the mechanisms that govern their collective activity remain to be understood. The connectome [12] provides detailed, quantitative information on connections between pairs of neurons in the entire organism, but anatomy alone—without physiology or dynamics—does not predict activity. Whole-brain imaging data [13,14] connects neuronal activity to behavior, yet does not explain how activity patterns are generated.

There is also a substantial theoretical literature, much of it consisting of phenomenological studies. With all of these available tools, we believe it is time to attempt a more systematic understanding of the neural mechanisms that drive function.

We propose that computational modeling may supply some of the missing information. In this work, we endeavor to build a model with the following properties: (i) The model should have a clear representation of anatomical structures and physiological measurements; without these components, model inferences are hard to interpret. (ii) The model should be as biophysical as possible. (iii) Dynamics being one of the missing ingredients, it would be desirable to have a dynamical systems model capable of producing the range of activity and subsequent behaviors observed. (iv) The model's responses must be realistic, i.e. its activity outputs must be similar to data. (v) Lastly, the model must be analyzable, as models are built to be interrogated.

We present in this paper a model of the *C. elegans* premotor network focusing on a core group of 10-20 neurons that experimentalists have identified as drivers of forward and reversal movements and model the dynamical interaction among them by a system of differential equations. The state variables are the neurons' calcium levels. This core group, which is small enough to be analyzable, receives inputs from over 100 presynaptic neurons. We do not model the dynamics of the presynaptic neurons. We instead treat their influence as an external force, i.e., we model premotor activity as a *driven dynamical system*. Leaving model details to the Results, we remark that we have leaned heavily on connectomic data for network architecture, and on many sets of whole-brain imaging data for model parameters.

A signature of *C. elegans* locomotion is its seemingly random switching between forward crawling and reversals. These switches are, in fact, far from random: they enable the *C. elegans* to forage for food and linger in a favorable environment. When provided with realistic input, our model can reproduce salient characteristics of switching dynamics. It does so through partial synchronization of clusters of neurons similar to the activity seen in the data. The dynamics produced are complex: they are neither completely random (as in the Markov chains models [5,15]) nor are they limit cycles (as in earlier dynamical systems models [16,17]). The close resemblance of our model outputs to data enhances the plausibility of our findings. Model analysis reveals, among other things, the differing roles played by gap junctions and synaptic connections in switching dynamics. It also enables us to identify specific neurons that are more influential.

## Previous models

We discuss below some models of *C. elegans* locomotion, focusing in particular on switching behavior. Many existing models fall into one of the following broad categories: *Phenomenological models* aim to describe or clarify empirical observations by connecting them to mathematical frameworks without claiming to reproduce biological mechanisms. *Mechanistic* and *data-driven models* make more direct connections to the anatomy and physiological data of the organism.

Several phenomenological models have been presented for the seemingly random switching between forward and reversal neurons, which underlies *C. elegans* locomotion. Ref [5] reproduced stochastic switching with a Markov model and proposed mutual inhibition and inherent stochasticity as the mechanisms for switching. Ref [15] proposed a model with random switches between discrete states, each described by linear dynamics. Chaotic heteroclinic networks can reproduce switching statistics, highlighting that deterministic neural dynamics can generate seemingly random state switches [18]. Ref [13] observed that motor commands are represented globally and found that the neural representation of motor sequences evolved on a low-dimensional manifold obtained from PCA. Ref [19] built a low-dimensional model of this neural activity actuated by control signals that induce switches. Ref [20] built a similar model using nonlinear dynamics. These models connect switching phenomena to mathematical frameworks, but it is hard to draw biological inferences from them, as they are too far removed from neuroanatomy.

In the second category are network models of premotor neurons in which the authors pose models containing biophysical parameters tied to cellular dynamics. Various model outputs are used to infer the values of model parameters.

Refs. [21] and [22] simulated the dynamics of a premotor and motor circuit. The stationary distributions of the motor neurons were then used to infer synaptic polarities, i.e., whether a synaptic connection is excitatory or inhibitory; there is little discussion of network dynamics. Ref [16] simulated the dynamics of the *C. elegans* neuronal network and evaluated the activity of the command neurons. Their model predicted that neural activity converges to limit cycles; i.e., all neurons eventually acquire the same periodicity. A number of other network models incorporating biophysical constraints assigned polarities to synapses through other means and explored the behavior of premotor neurons when activated by one or two specific sensory neurons [17,23,24]; Ref [17] also found that their network dynamics converged to a limit cycle.

A publication that appeared recently reproduces realistic *C. elegans* forward motion towards an attractor with a biomechanical body-environment model [25]. BAAIWorm is an integrative model that uses a neural network to control muscles that actuate the body-environment component of the model [25]. The neural network component is biophysically detailed with many parameters fit using machine learning and with an emphasis on its ability to produce coordinated forward movement.

Other recent work includes data-driven models introduced in Refs. [26] and [27]. Ref [26] applied independent component analysis with time-delay embedding to whole-brain imaging data to decompose the activity into meaningful components. Ref [27] assessed the extent to which the connectome captures signaling in the network and found that physical connections capture causal relationships.

The model we present in this paper is mechanistic and is constrained and fit with physiological, connectomic, and whole-brain imaging data. Our focus is on unraveling the dynamic mechanisms of switching behavior. We will discuss—after we present our results—the ways in which our results agree with or deviate from the findings in previous work.

## Results

We fit a dynamical systems model of premotor interneurons using several sources of data—voltage clamps, whole-brain calcium imaging, and the connectome. More precisely, we use the known connectome structure to set the locations of gap junction and synaptic connections and then use the whole-brain imaging data in conjunction with voltage clamp data to determine parameters, which consist primarily but not exclusively of the signs and weights of synaptic connections. Our model is a recurrent network of premotor neurons driven by a large collection of their presynaptic neurons; the presynaptic activity is also taken from the imaging data.

Our main results can be summarized as follows:

(1) Our model reproduces stochastic switching between the forward and reversal premotor neuron clusters, resembling data.
(2) The model reproduces neuronal time series whose higher-order statistics match those observed in the data.
(3) Using the deduced synaptic sign estimates, we identify presynaptic neurons that promote specific behaviors.
(4) We use our model to show how gap junctions and synaptic connections may collaborate to produce emergent stereotyped activity in the premotor network.

## Model description

To build a dynamical network of premotor neurons, we must first decide which neurons to include in the model. Experimental and whole-brain imaging studies have identified that various neurons are associated with forward and reversal locomotion, but there is no obvious way to isolate a subgraph of the connectome that corresponds to *the premotor network*. Premotor neurons receive inputs from many sensory neurons and interact with many other interneurons that may or may not be directly implicated in locomotion; these neurons, in turn, interact with a larger set of neurons, and so on. If all potentially relevant neurons were included, the resulting network would be exceedingly large; see Fig 1A. Such a model would be impossible to fit due to a deficit of labeled neurons in whole-brain imaging data and difficult to analyze due to a large number of variables and parameters.

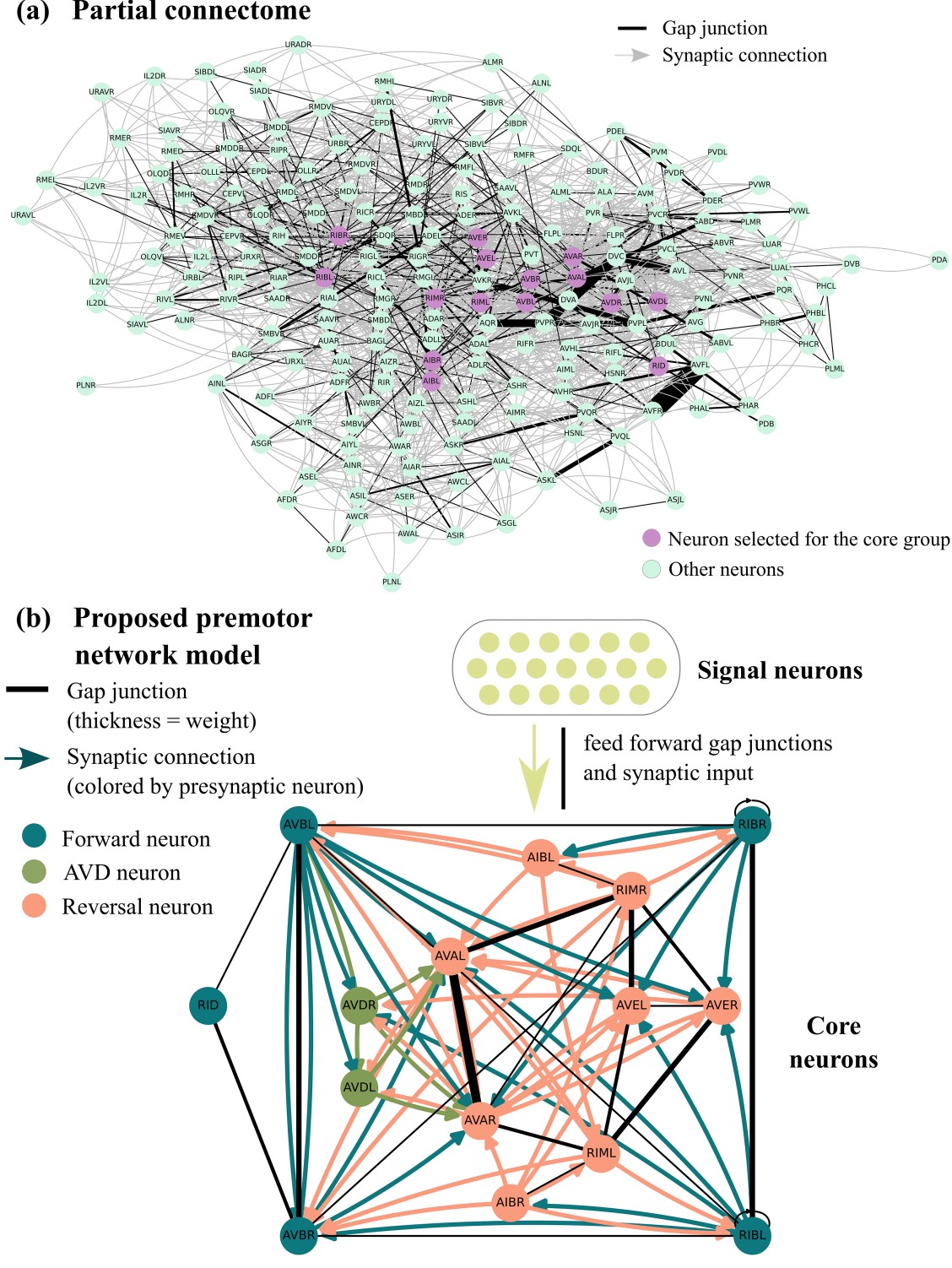

**Fig 1**. **(a) Partial connectome from [12] containing 198 sensory neurons, interneurons, and motor neurons (most motor neurons that form neuromuscular junctions with muscle cells are excluded).** The 15 neurons that will be selected as core neurons are shown in purple. Additionally, the partial connectome contains 137 neurons that are directly connected to the core neurons either presynaptically or through gap junctions. (b) Core neurons selected for the premotor network model and the connections among them. We categorize most core neurons as either forward or reversal based on the connectome and previous experimental work.

Attempting to strike a balance between biological realism and analyzability, we propose a model with the following basic structure: a core group of neurons whose dynamics we simulate while influenced by a larger group of neurons whose activity is applied as input. More precisely, neurons in our network are divided into two groups. The first is a core group of neurons deemed the most relevant to the study in question. For us, these would consist of interneurons believed to play the most important roles in *C. elegans* locomotion. Neurons in this group are coupled to one another in a recurrent network; we will refer to them as *core neurons* for simplicity. This group should be relatively modest in size, though as large as need be. Then there is a second group consisting of what we will refer to as *signal neurons*. Signal neurons provide feedforward input to the core group; we do not model their dynamics, gleaning their outputs instead from various data sets. Core neurons are assumed to only receive input from other core neurons and signal neurons.

Mathematically, we build a model in the form of a *driven dynamical system*, defined by a set of ordinary differential equations describing the dynamics of neurons in the core group, together with a time-dependent forcing representing the inputs from signal neurons. Below we identify the neurons to be modeled and give the equations governing model dynamics.

**Selection of neurons.** Guided by connectomic, whole-brain imaging, and experimental data, and constrained by data availability, [6,9,12–14], we converge to a set of 15 neurons that, we posit, play central roles in forward-reversal locomotion. These are the neurons we place in the core group in our model. Below we recall some basic biological facts about them, and explain why they were chosen.

Conceptually, we cluster the 15 core neurons into three categories: (i) *Forward neurons* (AVBL, AVBR, RIBL, RIBR, and RID); the AVB neurons are known to be command premotor neurons that drive forward movement; they are connected to the other interneurons in the group via gap junctions [5,6,9,12,22,28–30]. (ii) *Reversal neurons* (AVAL, AVAR, RIML, RIMR, AVEL, AVER, AIBL, and AIBR); the AVA neurons are known to be command premotor neurons that drive reversal movements; they are connected to the other interneurons in the group via gap junctions [16,31]. The forward and reversal clusters above have many synaptic connections within and between clusters. Finally, (iii) there are two neurons (AVDL and AVDR), which are highly interconnected with the two clusters via synaptic connections. A diagram showing these 15 neurons together with their known connections is shown in Fig 1B.

According to the connectome, 137 neurons provide input to the core group [12]; 112 of the 137 input neurons appear in the whole-brain imaging data used to fit our model [14]—these are the 112 signal neurons defined in our model. Divergent from our model, in the real *C. elegans* brain the 15 core neurons have many recurrent connections with the signal neurons—we have chosen to neglect these recurrent interactions in favor of a more tractable dynamical system.

We remark that though guided by data, there is some arbitrariness in the choices above as well as practical constraints. Most notably, we do not include the PVC and DVA neurons in our core set despite their strong association with locomotion and strong connections with the core set. This is because the PVC and DVA neurons do not appear in the whole-brain imaging datasets we use to fit our model [14], making it infeasible to fit model parameters for these neurons using the methods we employ. Another omission of note is that we exclude most motor neurons, many of which provide feedback to our core group via gap junctions [12]. As these neurons are "downstream" from our network, we do not wish to model them as signal neurons.

**Model dynamics and parameters.** We begin with model equations for a generic set of $n$ core neurons and $m$ signal neurons. Conditions having to do with the specific core neurons selected are imposed only at the end.

As *C. elegans* neurons have graded responses to injected currents, it is natural to measure their activity in terms of calcium levels [32]. We set the state variable for each neuron to be its GCaMP z-score; these measurements are acquired from whole-brain imaging data [14]. Let $x_1, \cdots, x_n$ denote these scores for the $n$ core neurons and $x_{n+1}, \cdots, x_{n+m}$ the scores

for the signal neurons. We posit that for $i \leq n$, the time evolution of $x_i$ is given by

$$\tau \frac{dx_i}{dt} = f_i(x_i) + \beta \sum_{j=1}^{n+m} \mathbf{W}_{ij}(x_j - x_i) + \sum_{j=1}^{n+m} \mathbf{A}_{ij}\sigma(x_j). \tag{1}$$

The first term on the right side describes neuron $i$'s intrinsic dynamics and the second term represents the combined influence from all core and signal neurons $j$ that form gap junctions with neuron $i$. The third represents the influence from core and signal neurons presynaptic to neuron $i$; $\tau, \beta, \mathbf{W}_{ij}, \mathbf{A}_{ij}$ are constants, and $\sigma() = \text{ReLU}()$ is the ReLU activation function.

As stated earlier, the dynamics of $x_i$ for $i > n$, i.e., for signal neurons, will not be modeled. Instead, $\{x_i(t)\}$ is extracted from whole-brain imaging time series.

The intrinsic dynamics of individual neurons are approximated by voltage clamp data [32,33]. Leaving details to the Methods, we assume $f_i(x_i)$ has the form

$$f_i(x_i) = -2(x_i + 0.8)(x_i - 0.1)(x_i - 1) + \mathbf{d}_i$$

where the bias term $\mathbf{d}_i$ is different for each neuron. In the absence of input from other neurons and without a bias term, the system has two stable states, a low-activity state at $x_i = -0.8$ and a high-activity state at $x_i = 1$. A positive bias $\mathbf{d}_i$ makes the neuron's high-activity state relatively more stable while a negative $\mathbf{d}_i$ makes the neuron's low-activity state more stable. The cubic form of the neurons' intrinsic dynamics approximates the typical nonlinear response of *C. elegans* neurons to current injection [32,33].

The influence from both gap junctions and synaptic connections is approximated as linear, but the latter involves a rectifier activation function. The rectifier activation function applied to the neural activity of the presynaptic neuron reflects our assumption that neurons influence their postsynaptic counterparts synaptically only when depolarized and do not affect postsynaptic neurons when hyperpolarized.

The quantities that remain to be determined are $\mathbf{d}_i$ (bias in the intrinsic dynamics of individual neurons), $\mathbf{W}_{ij}$ (relative gap junction weights between neurons), $\mathbf{A}_{ij}$ (synaptic weights), $\beta$ (relative contribution of gap juction input), and $\tau$ (timescale parameter). These quantities depend on the specific neurons in the model, and here is how they are determined:

The gap junction weights $\mathbf{W}_{ij}$ are derived from connectome data [12] (see Methods); for the neurons in our model, they are shown in Fig 2A.

As for the synaptic weights $\mathbf{A}_{ij}$, we set $\mathbf{A}_{ij} = 0$ when the connectome shows no connection between the relevant neurons; absent connections are depicted by white boxes in Fig 2B. When a connection is present, connectome data offer information on the number of synapses *without sign specification* [12]. Interpretation of this data is further complicated by the fact that many synaptic connections within the locomotion subnetwork are known to be "complex", meaning the connection contains both excitatory and inhibitory neurotransmitter-receptor pairs [34]. In the absence of clear guidance, we have elected to estimate independently the values of $\mathbf{A}_{ij}$, which in our model can be positive or negative representing the *net* excitatory/inhibitory inputs.

The $\mathbf{A}_{ij}$ not set to 0 *a priori*, the values of $\mathbf{d}_i$, and $\beta$ are estimated by linear regression. We first find datasets from [14] that contain a sufficient quantity of core neurons, resulting in 22 datasets that we use to fit our model. We then perform a parameter sweep for $\beta$ over many linear regressions. The minimum combined error occurs when $\beta = 10$ (See Methods). After setting $\beta$ to be 10, we perform separate linear regressions for each of the 22 datasets and average over all of the dataset parameters to obtain a single general set of parameter values $\mathbf{A}_{ij}$ and $\mathbf{d}_i$. The obtained values for the synaptic weights and signs, $\mathbf{A}_{ij}$, are shown in Fig 2B.

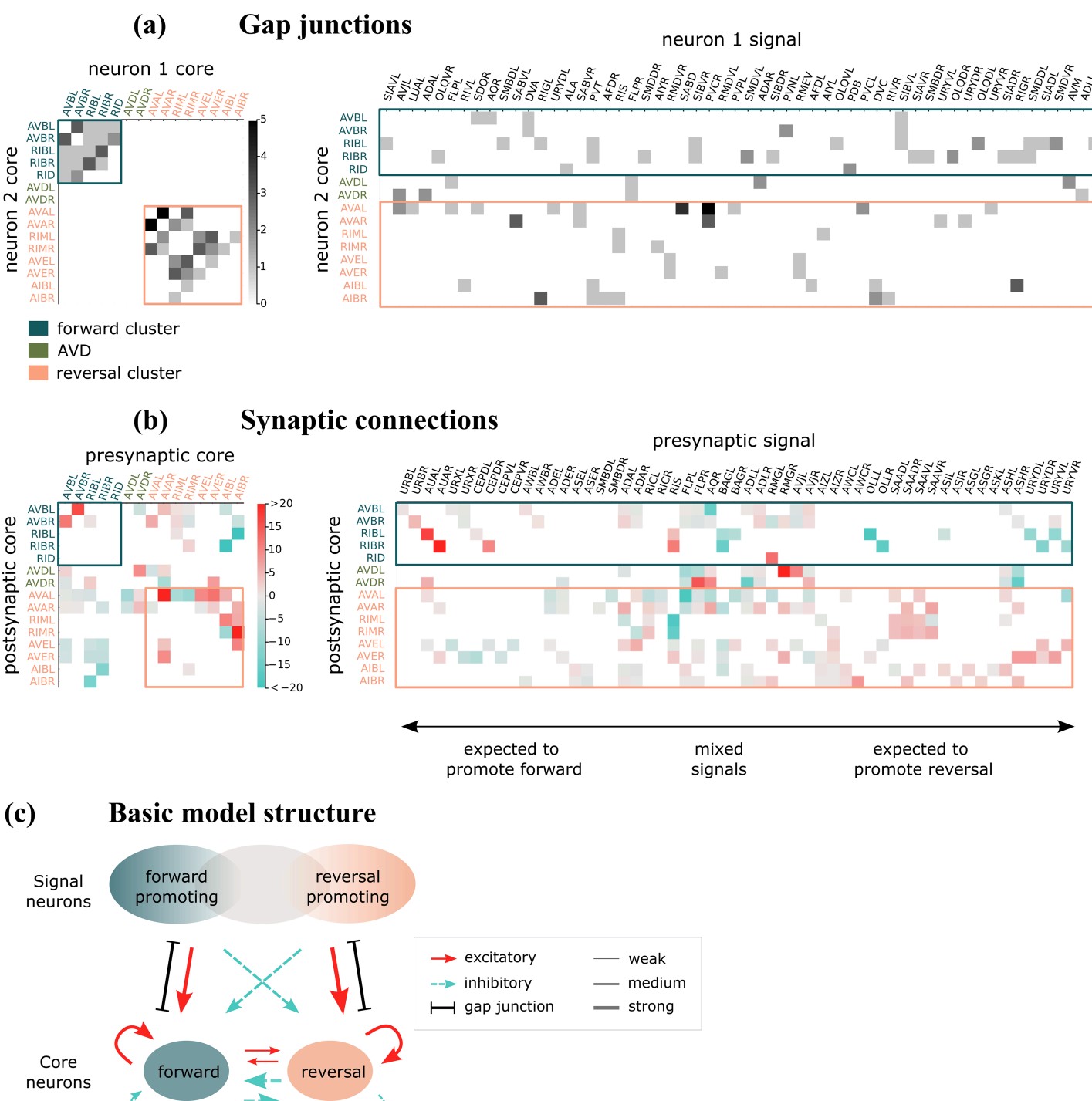

**Fig 2**. (a) Relative gap junction weights between core neurons and between signal neurons and core neurons (select signal neurons are shown). Weights are taken directly from [12]. (b) Synaptic signed weights between core neurons and between signal neurons and core neurons derived from regression (select signal neurons are shown). Signal neurons are loosely sorted as forward-promoting or reversal-promoting. (c) Cartoon of principal structure between signal neurons, forward core neurons, and reversal core neurons.

The last parameter that must be fit is the timescale parameter $\tau$. We fit $\tau$ by simulating the activity of the core neurons in the six datasets shown in Fig 14 for variable $\tau$ values. We select $\tau$ such that, on average, the simulated core neurons switch at the same speed as the real neurons (see Methods, Figs 13 and 14).

Recapitulating, the neurons in our core group contain two subsets—the "forward cluster" and the "reversal cluster"—that are known from experiments to be associated with forward and reversal locomotion. (See the paragraph on neuron selection). Simulated dynamics of these core neurons will be studied and compared to data. We note that in our model, dynamical properties of individual neurons are emergent; the only information specific to a neuron that we have built into the model are gap junction weights and the presence or absence of synaptic connections.

## Model validation

In this section we compare model outputs to data, demonstrating that though much simplified, our model can reproduce many key features of *C. elegans* locomotion.

**Switching dynamics.** A salient characteristic of *C. elegans* locomotion is a constant switching between forward and reversal movements at characteristic yet variable time intervals. An appropriate first test of our premotor network model is thus to evaluate if the model can reproduce these switching dynamics. In Fig 3 we compare premotor neuron activity from dataset 2023-01-23-15.json [14] to simulated activity of the corresponding neurons. Of interest are the activity levels of neurons in the core group; $x_i(t)$ from this dataset will be compared to simulation results from our model when driven by signal neurons whose outputs are taken from the same dataset. Fig 3A shows $x_i(t)$ for individual neurons over a 12-minute duration: the top panel is data from [14], the bottom panel is the simulation results. In both cases, neurons in the forward and reversal clusters, and the AVD neurons, are shown in three different colors, and the partial synchronization within each group is visible. Qualitatively, the forward and reversal clusters in the simulation switch between high and low states similar to the switches seen in the data.

Assuming that a high state of the forward cluster represents forward locomotion and the analogous statement for the reversal cluster, Fig 3B shows the forward/reversal direction of movement for this *C. elegans* as a function of time. We let $F(t)$ be $x_i(t)$ averaged among the forward neurons and let $R(t)$ be $x_i(t)$ averaged among the reversal neurons. We assume that when $F(t) - R(t) > 0.5$ (dark green) the *C elegans* is engaging in forward movement as, on average, the forward neurons are more active than the reversal neurons. When $F(t) - R(t) < -0.5$ (tan) we assume it is in reversal as the reversal neurons are more active. For $|F(t) - R(t)| < 0.5$ (shown in grey), we consider the movement to be ambiguous or paused. A comparison of the true and inferred locomotion state is shown in the Appendix. The two strips showing data and simulations are remarkably similar given the many simplifying assumptions made in the design of our premotor network model (Fig 3B). More examples comparing data to simulations are shown in Fig 14.

**Higher order statistics.** Fig 3C shows the correlations between pairs of neurons. Data and simulation compare well: Neurons in the forward cluster are highly correlated and anticorrelated to the reversal neurons. An analogous statement holds for neurons in the reversal cluster. The AVD neurons are not strongly correlated with either cluster in both the data and simulation.

Fig 3D shows the probability distribution functions of the activity levels for the neurons in question. Here as well, model statistics resemble data: e.g. the forward neurons are bimodal in their pdfs and the reversal neurons have a single strong peak below zero and a large positive tail. The distributions of some simulated neurons fit the data better than others. For example, the RID neuron has a larger reversal peak than forward peak in the data but the opposite is true in the simulation. The KL divergence is a metric that could be used to quantify distribution differences and used in a loss function to fit a model with simulation distributions [35].

Fig 3E shows the dwell time distributions for the forward and reversal states. The histograms are qualitatively similar, with mean dwell times between 0.2 and 0.33 minutes in both data and simulations. Simulated dwell times are slightly longer, but note that the time constant $\tau$ in our model is optimized over all data sets.

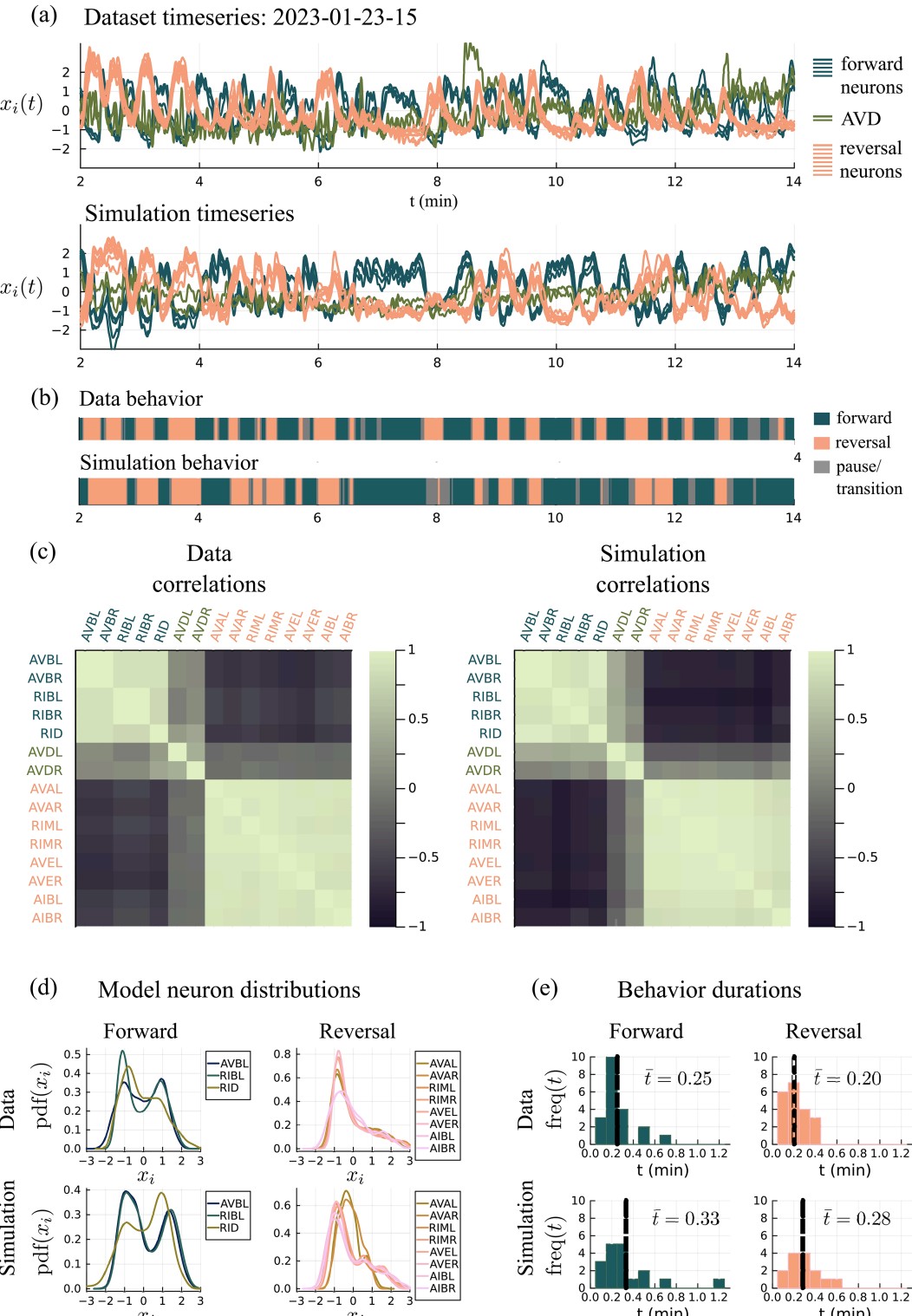

**Fig 3**. **Comparison of data and model.** (a) Core neuron activity in the dataset 2023-01-23-15 [14] versus the simulation (101 neurons labeled). (b) The dominant cluster of neurons—forward or reversal—is a proxy for behavior (see Appendix). (c) Pairwise correlations between neurons. (d) Distribution of neural activity. (e) Dwell times in the forward and reversal states. Histograms are generated from the forward/reversal sequences in (b) using bin size 0.1 and excluding dwell times less than 0.05 min.

We stress that the parameters in our network model are deduced from many datasets and are not specific to the dataset 2023-01-23-15. Indeed, our model simulates forward/reversal switching that resembles the actual switching in many datasets (Fig 14).

### Analysis

**Gap junctions versus synaptic connections.** Next, we use our premotor network model to examine the roles played by gap junctions versus synapses in the stochastic switching between forward and reversal seen in Fig 3. First, we observe two salient characteristics in the dynamics of the forward neurons. Fig 4A shows the time series from a data set. Fig 4B shows the corresponding time series, simulated using our premotor network model when driven by signal neurons from the same dataset. Note that in the data as in the model, the forward neurons are synchronized, and together they make irregular but characteristic switches between high and low states.

Next, we study the effect of removing the synapses in the model, leaving only gap junctions. Fig 4C shows that the forward neurons stay correlated but fail to produce clear, strong switches between the high and low states. When gap junctions are removed leaving only synaptic interaction, the forward neurons exhibit dramatic switches between states but do not stay synchronized (Fig 4D).

Similar results are observed for the reversal cluster (Fig 4E–4H).

These results show that in our premotor network, both gap junctions and synaptic connections are needed to produce the switching behavior observed in data. They play different roles: gap junctions synchronize, while the stochastic switching appears to be mediated by synaptic dynamics. We hypothesize that similar mechanisms may be at work in the real *C. elegans* brain.

**Promoter and suppressor signal neurons.** Some signal neurons are more influential than others in promoting a switch between behavioral states. We ranked the signal neurons in dataset 2023-01-05-01 [14] by the extent to which they excite or inhibit the core forward or reversal neurons through synaptic connections (Fig 5). We then evaluated the impact of setting the top promoters or suppressors to zero, in analogy with ablation experiments.

A switch from reversal to forward could in theory be generated by either exciting the forward neurons or inhibiting the reversal neurons. The top forward promoters (as measured by their cumulative excitatory input to the forward set preceding a switch to forward) are RMED, AUAL, and CEPDR (Fig 5A). Neutralizing these neurons results in extended reversal intervals and shortened forward intervals; the fraction of time spent in the forward state decreased from 0.38 to 0.14 while the fraction of time spent in the reversal state increased from 0.34 to 0.59, indicating that these neurons promote forward locomotion considerably. The top reversal suppressors (as measured by their cumulative inhibitory input to the reversal set preceding a switch to forward) are FLPL, CEPDL, and URXR (Fig 5B). In contrast to the forward promoters, setting the top reversal suppressors to zero does not have a significant impact on behavior; see Fig 5B.

A switch from forward to reversal could be generated by either exciting the reversal cluster or by inhibiting the forward cluster. The top reversal promoters are URYVL, URYDR, and AIZR while the top forward suppressors are OLLL, URYVL, and URYDR; the URY neurons simultaneously excite the reversal neurons and inhibit the forward neurons in the model (Fig 5C–5D). Setting the top three reversal promoters to zero increases the fraction of time spent in the forward state from 0.38 to 0.57 and reduces the fraction of time spent in the reversal state from 0.34 to 0.17. Similar changes are observed when setting the top three forward suppressors to zero.

In summary, we deduce from the analysis above that the drivers of forward and reversal motions are not symmetric: while forward motion requires stimulation of the forward cluster, reversal can be driven by either promotion of reversal neurons or suppression of forward neurons, suggesting that reversal may be a braking mechanism for forward motion.

Analysis for another dataset is shown in the Appendix (Fig 15).

## Forward neuron cluster

(a) Forward neurons: dataset 2023-01-23-15

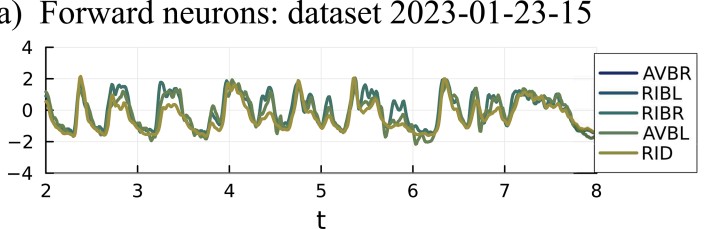

(b) Simulation: gap junctions and synapses

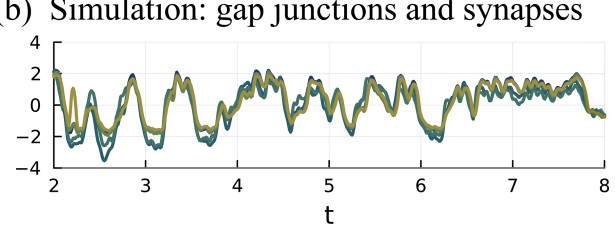

(c) Simulation: only gap junctions

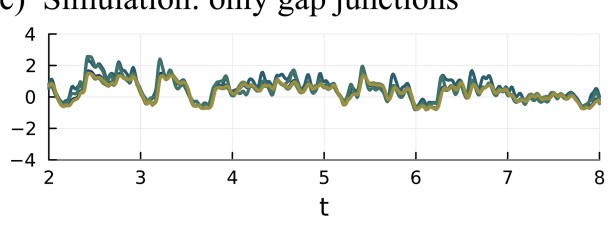

(d) Simulation: only synapses

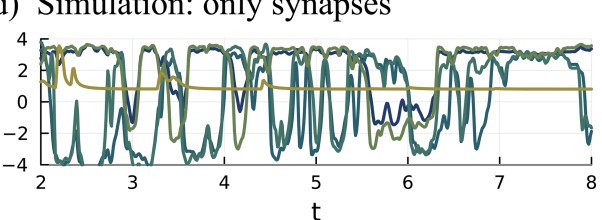

## Reversal neuron cluster

(e) Reversal neurons: dataset 2023-01-23-15

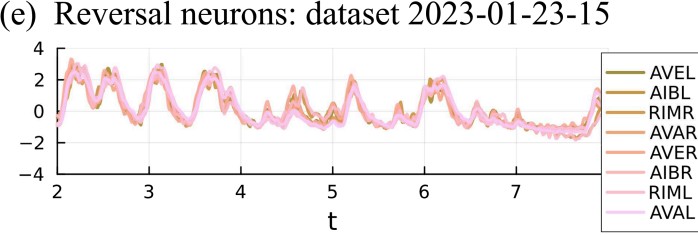

(f) Simulation: gap junctions and synapses

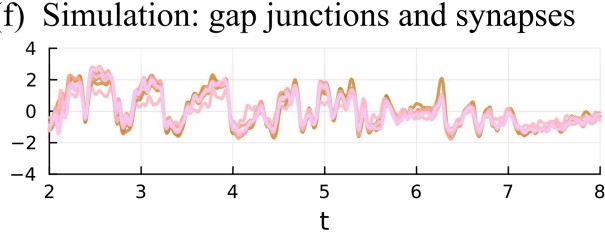

(g) Simulation: only gap junctions

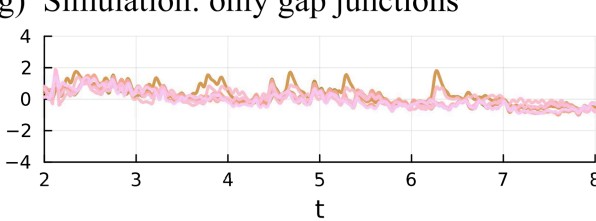

(h) Simulation: only synapses

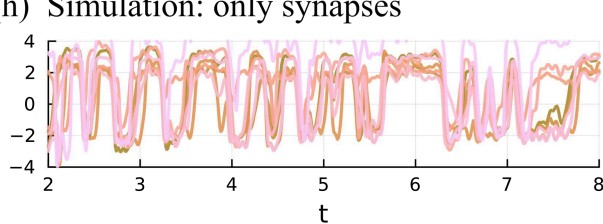

**Fig 4**. **Forward and reversal cluster dynamics using connected neurons as signals.** (a) Forward neuron activity from dataset 2023-01-23-15. (b) Simulated forward neuron activity using both gap junctions and synapses. (c) Simulated forward neuron activity using only gap junctions. (d) Simulated forward neuron activity using only synapses. (e) Reversal neuron activity from dataset 2023-01-23-15. (f) Simulated reversal neuron activity using both gap junctions and synapses. (g) Simulated reversal neuron activity using only gap junctions. (h) Simulated reversal neuron activity using only synapses.

**Ablation of core neurons.** The time series of the forward neurons are highly correlated, as are the reversal neurons (Fig 3), yet the connectivity of the core set is not uniform—each neuron forms a unique set of connections with other neurons in the network. It is unclear how each neuron within the core set contributes to the switching dynamics of the core network. To evaluate the contribution of each neuron class we perform simulations with ablations to the core set.

Fig 6A shows the simulated baseline activity using signal neurons from dataset 2023-01-23-15 [14] with no ablations. We ablated (severed) both the synaptic connections and gap junctions in the model between each core neuron class and the remainder of the network and observed the impact that this has on the core network dynamics.

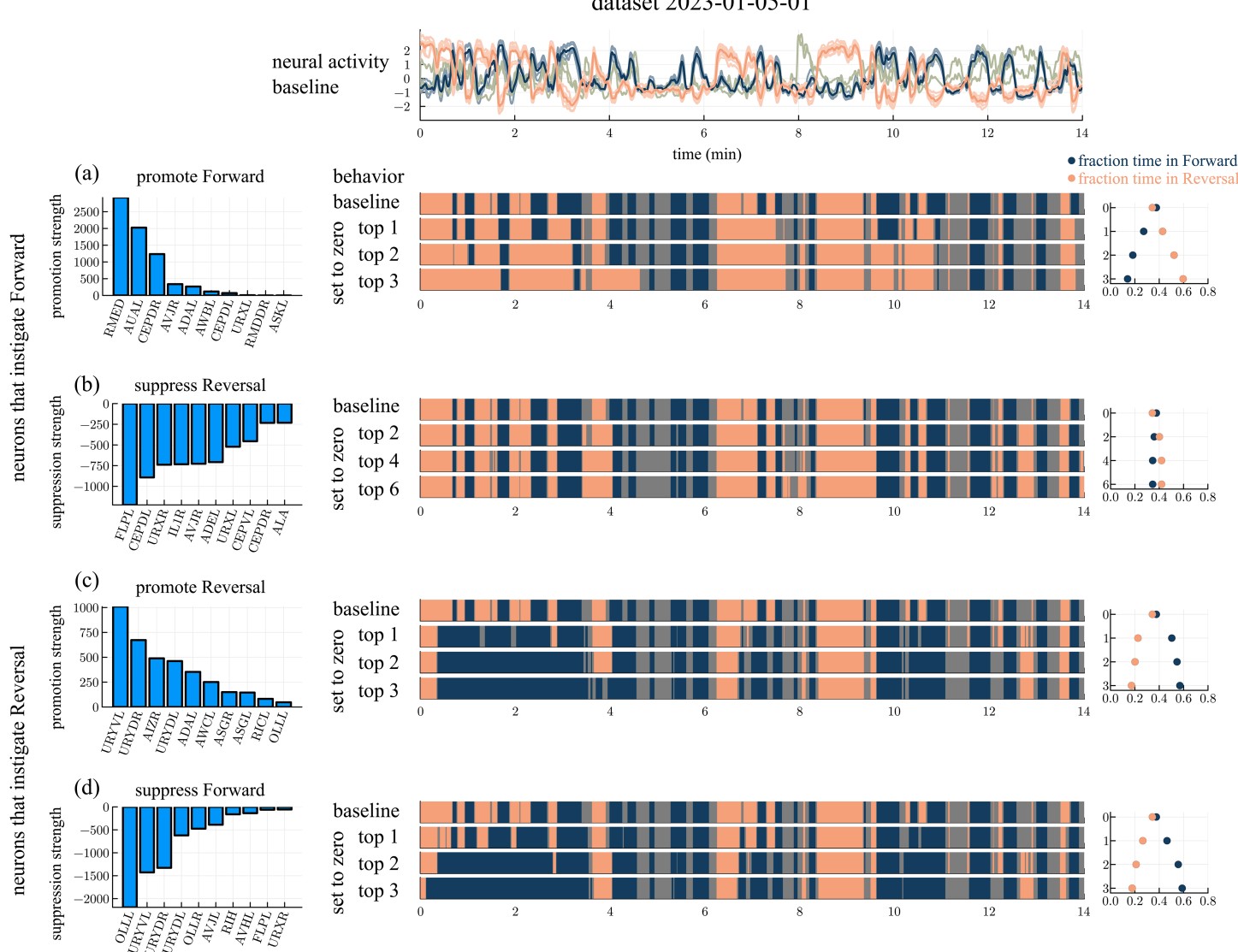

**Fig 5.** **Simulations using signal neurons from dataset 2023-01-05-01 [14].** Promoter and suppressor neurons from dataset 2023-01-05-01 [14] are ranked by the strength of their promotion or suppression (left column). Top promotors/suppressors are then set to zero, and resulting activity strips are shown (middle column). Fractions of time spent in forward and reversal motion are summarized (right column). (a) Signal neurons that promote the forward neurons. Behavioral time series in data (row 1), with the top promoters set to zero (rows 2,3,4). Analogous results are presented in (b), (c) and (d) for signal neurons that suppress reversal, promote reversal, and suppress forward neurons respectively.

Among the forward neurons, ablating the RIB neurons had the largest impact on the dynamics resulting in significantly reduced forward intervals (Fig 6B). Ref [36] found that ablating RIB neurons reduced locomotion speed but did not significantly impact reversal frequency. In our model, such an ablation resulted in the conversion of a fraction of time spent in the forward state to the paused state, consistent with a slow-down of the nematode. Precisely, it reduced the fraction of time spent in the forward state from 0.46 to 0.23 and increased the fraction of time spent in the pause state from 0.13 to 0.44 (Fig 6B).

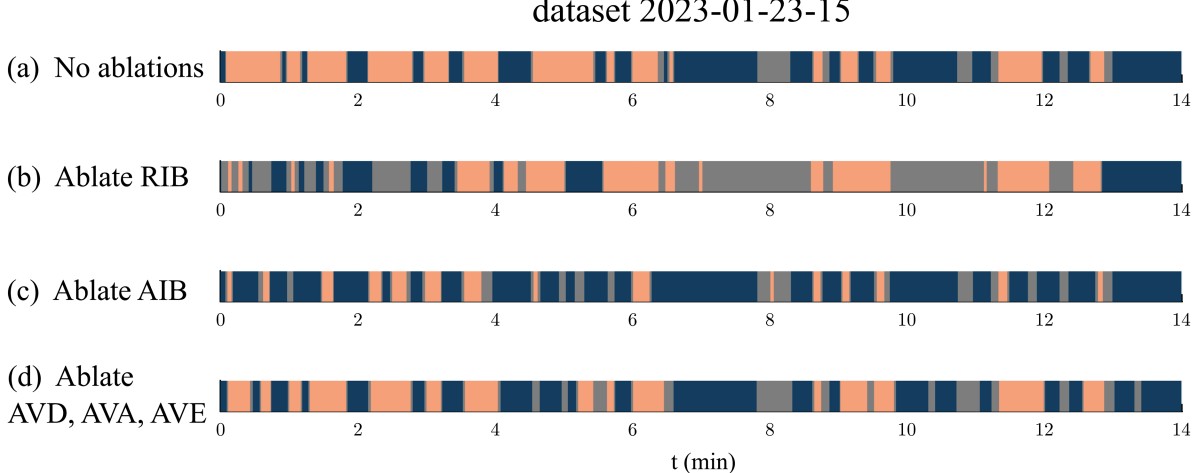

**Fig 6**. **(a) Core neurons simulation with no ablations, using dataset 2023-01-23-15 for the signal neurons.** (b) Core neurons simulation with RIB neurons ablated. (c) Simulation with AIB neurons ablated. (d) Simulation with AVD, AVA, and AVE neurons ablated. In all simulations, forward and reversal intervals of less than 2 seconds dividing a pause state were set to a pause state. Pause intervals of less than 5 seconds dividing a forward or reversal interval where set to the enclosing locomotive state (F or R).

Among the reversal neurons, ablating the AIB neurons had the largest impact on the dynamics resulting in reduced reversal intervals and extended forward intervals (Fig 6C). Ref [37] found that experimental ablations to the AIB neurons resulted in extended forward movement durations. In alignment with this finding, ablating the AIB neurons in the model increased the average forward duration time from 0.30 to 0.38 and reduced the average reversal duration time from 0.39 to 0.14. Ref [38] found that ablations to the AIB neurons suppressed reversal frequency, a phenomenon that did not occur in the model.

Ref [38] also found that ablating the AVD, AVA, and AVE neurons combined resulted in a similar amount of reversal frequency suppression as ablating the AIB neurons alone. Ablating the AVD, AVA, and AVE classes combined in the model had a smaller impact on the dynamics than ablating the AIB neurons alone, in agreement with Ref [38] and further highlighting the importance of AIB neurons for reversals (Fig 6D).

**Signal propagation maps.**  One way to investigate the sign of a neural connection, i.e., whether it is excitatory or inhibitory, is through optogenetic stimulation. Ref [39] applied optogenetic stimulation to *C. elegans* neurons one at a time and then measured the response in other neurons. Fig 7A shows the signal propagation map for the core neuron set, reproduced from Ref [39] data. Each column shows the strength of the post-stimulus neural activity in response to stimulation applied to the neuron at the top of each column.

From simulated data, we generate an analogous signal propagation map (Fig 7B). Core neurons are stimulated one at a time without any other signal input, and network responses are simulated. Post-stimulus displacement for a target neuron is computed by measuring its steady state activity level during stimulation and subtracting from that the neuron's baseline value (when unstimulated). In the absence of any stimulus, the neurons converged to a fixed point of the undriven core neuron dynamics. The forward neurons stabilized at higher values than the reversal neurons, indicating that the core network contains bias in favor of forward locomotion. Ref [40] also found that the *C. elegans* neuronal network is biased in favor of forward-circuit activity.

**(a)** Randi et al. (2023) signal propagation

**(b)** Post-stimulus average displacement

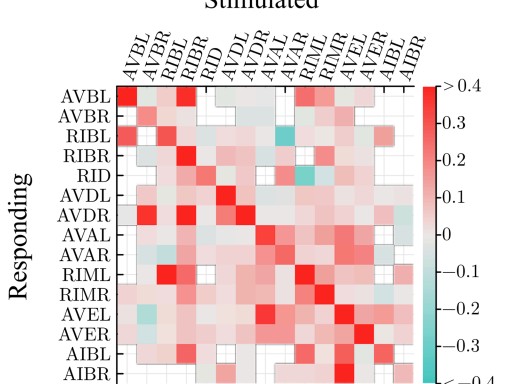

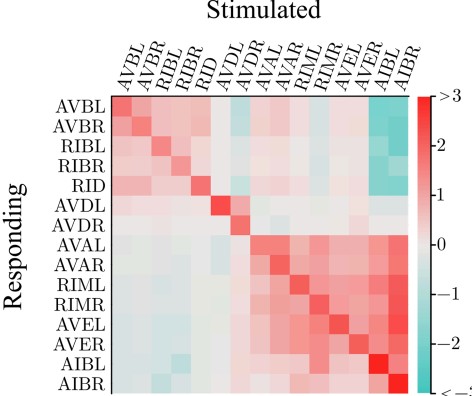

**Fig 7.** **(a) Signal propagation map reproduced from Ref [39] data.** (b) Post-stimulus average displacement computed from the simulation displacement trajectories.

The signal propagation map generated from the model shares some similarities with the map in Ref [39] but differs from it in several ways: Excitatory responses within the forward and reversal clusters are stronger and less varied than corresponding responses in the experimental map. In the model, forward neuron perturbations consistently inhibited the reversal neurons while in the experiment they both excited and inhibited the reversal neurons depending on the connection.

A possible explanation for these discrepancies is the existence of pathways not represented in the model. Specifically, two neurons in the core group can be—in fact likely are—connected by multiple pathways that involve neurons from outside of the core group. When multiple pathways are involved, the summed response tends to be nuanced, consistent with what is seen in the experimental map. The exclusion of such pathways simplified the model substantially but as this analysis shows, it is also a limitation.

Another implication of these additional pathways is that while the functional relationship between two neurons is good indication of the sign of the synapse between them, one cannot take for granted that the two necessarily coincide. Other factors that impact the functional relationship are gap junctions, intrinsic dynamics, and extra-synaptic signaling—these additional factors can also make the functional relationship sign deviate from the synaptic sign.

**Replicating the impact of heat shocks.** External stimulus impacts the neural dynamics of *C. elegans* and can change how behavior is encoded [14]. Ref [14] found that applying a heat shock to the nematode increased its reversal probability post heat shock (Fig 8B) [14]. The authors also identified several heat responsive neuron classes: ADA, AVD, AWB, and AWC exhibited excitatory responses to heat shocks and neuron classes AIM, OLQD, OLL, and AIY exhibited inhibitory responses (Fig 8A).

We mimicked the impact of heat shocks to model neurons by directly applying positive perturbations to signal neurons ADAL, AWBL, and AWCL and negative perturbations to signal neurons OLQDL, OLQDR, OLLL, and OLLR in dataset 2023-01-05-01 (Fig 8C). Perturbation shapes were chosen to emulate the responses of these neurons in data (see Fig 8A).

Fig 8D shows the fraction of model simulations in a reversal state immediately before and after the neural perturbations above were applied, averaged over 44 perturbation events. The fraction of reversals was significantly increased in alignment with data. However, unlike data, the model did not show a persistent state that lasted beyond the end of the

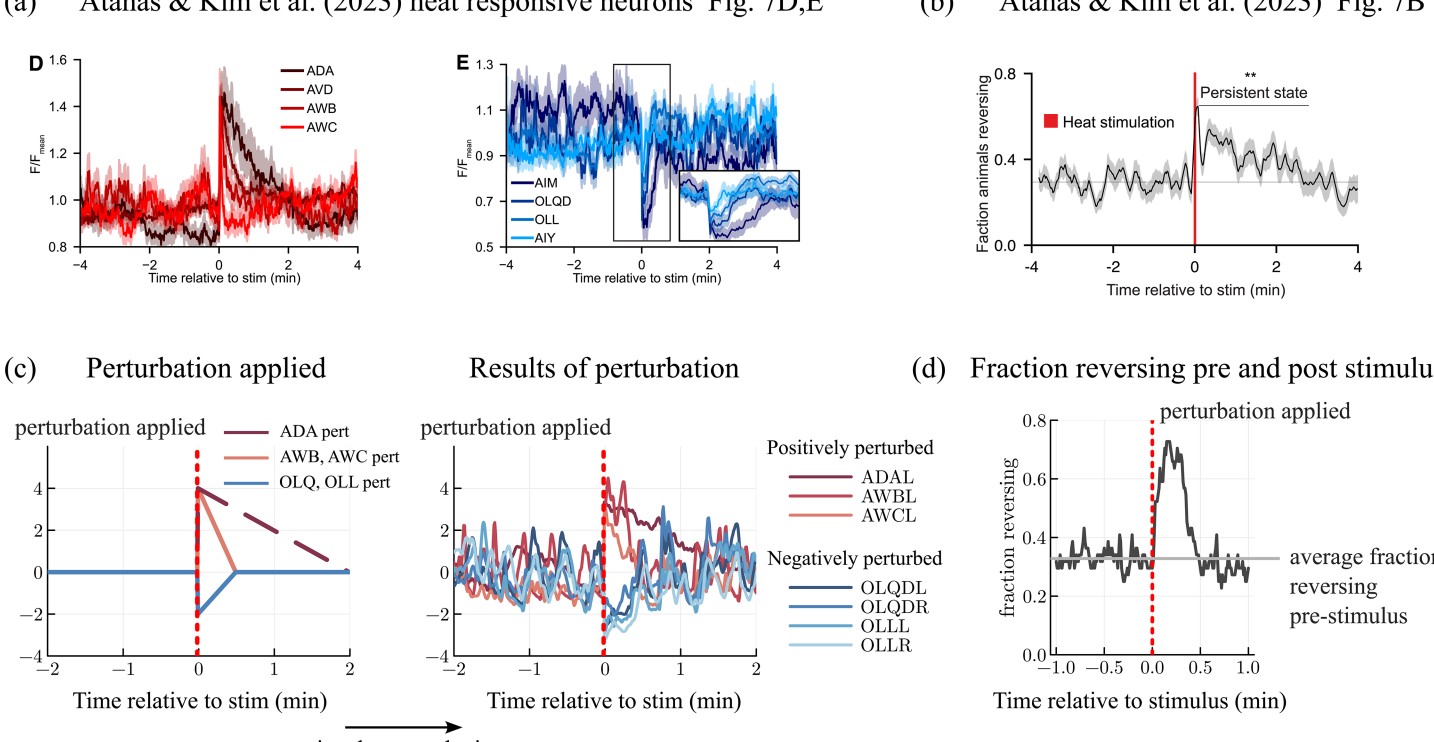

**Fig 8.** **(a) Event-triggered averages of neural activity aligned to the heat stimulus for some neurons with (D) excitatory or (E) inhibitory responses to the stimulus [14].** (b) Event-triggered averages of behavior of 32 animals in response to the heat stimulus [14]. (a,b: Reprinted from *Cell*, 186(19), Atanas, A. A. & Kim, J., et al., Brain-wide representations of behavior spanning multiple timescales and states in C. elegans, 4134-4151, Copyright (2023), with permission from Elsevier.) (c) Positive and negative perturbations applied to heat responsive neurons and resulting time series of heat responsive neurons with the perturbation applied. (d) Fraction reversing before and after perturbations are applied, averaged over 44 perturbation events.

perturbation. This should not be surprising as the neurons we perturbed are far from the only ones impacted by a real heat shock.

**Behavior over long durations: Roaming *vs* dwelling.** While semi-regular switching between forward and reversal is characteristic of *C. elegans* locomotion, the patterns and frequencies of these switches are known to convey information on the behavior of the worm over longer time durations. Roaming and dwelling are two such complementary behavioral states. Roaming allows *C. elegans* to explore a large space to find food while dwelling keeps *C. elegans* in a more confined region so that it can, e.g., exploit a discovered food source [30]. Roaming is characterized by long stretches of forward runs interrupted by infrequent reversals while dwelling is characterized by shorter forward runs and frequent, short, reversals [30].

Using different datasets as signal input, the model produces various behaviors over time durations of 10-15 minutes. Fig 9 contrasts a behavioral sequence that is indicative of A dwelling behavior and B roaming behavior using signal input from two Ref [14] datasets: 2023-01-09-22 and 2023-01-23-15. From the behavioral time series of forward/reversal transitions we construct locomotory paths that simulate *C. elegans* locomotion on an agar plate: Following [41], a simplified pattern of forward-run, reversal, turn, and resume-forward is imposed. The durations of forward and reversal are given by

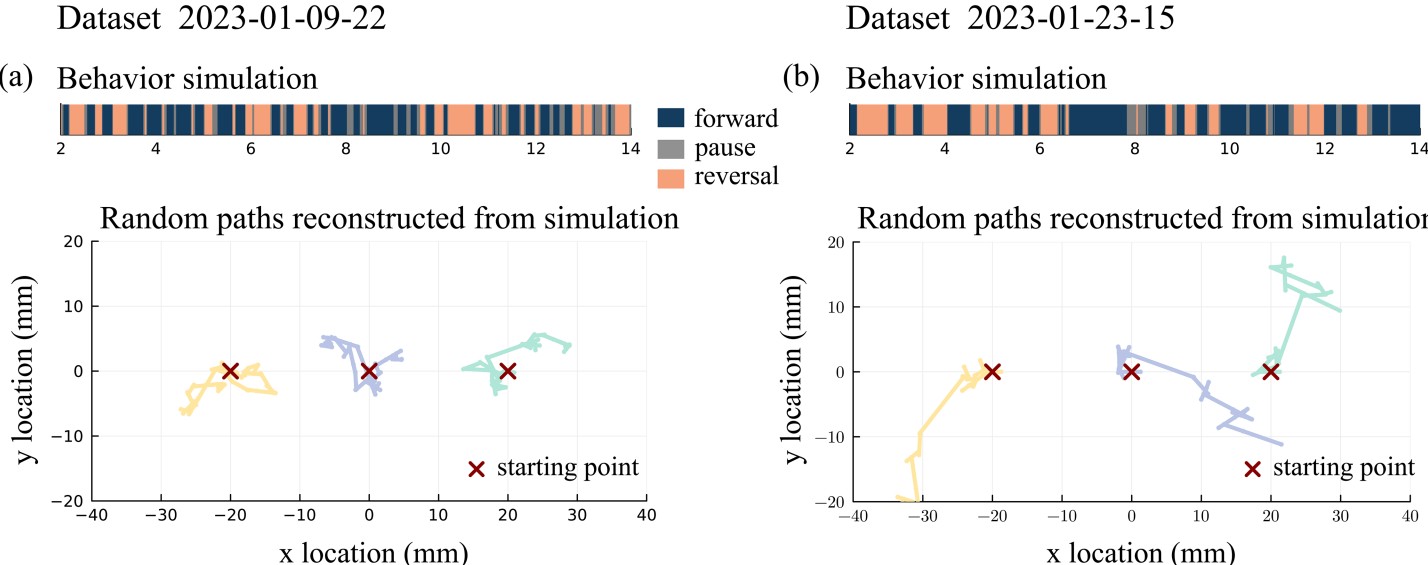

**Fig 9.** (a) Behavioral state time series from the model simulation using dataset 2023-01-09-22 [14], and three locomotion paths simulated from this behavioral time series. The simulation procedure is described in the Methods. (b) Analogous results using dataset 2023-01-23-15 as input for the simulation.

the behavior simulation while forward and reversal speeds are fixed and turn angles are randomly selected from a distribution. Different draws of turn angles result in different realizations of the locomotory path, three of which are shown for each of the two datasets (Fig 9). More details on path reconstruction are given in the Methods.

Signal input from dataset 2023-01-09-22 produces high frequency switches, resulting in shorter forward runs and reversals (Fig 9A). As a result, the *C. elegans* stays confined to a smaller region over the 12-minute time interval; this behavior is characteristic of dwelling. Signal input from dataset 2023-01-23-15 produces longer forward runs and longer reversals; the resulting paths in Fig 9B explore larger regions than the paths in Fig 9A. These paths show characteristics in the direction of roaming.

## Discussion

### Relation to existing literature

The functions of individual *C. elegans* neurons and their association with behavior have been extensively studied [13, 16,30,42–44]. Experimentalists have identified interneurons that are primarily responsible for driving and facilitating the primary locomotion behaviors of forward crawling, reversals, and turns [9,37,38]. Ablation studies show that key premotor neurons are essential for producing forward and reversal movements [45,46]. Whole-brain imaging shows that many neurons beyond the command premotor neurons are correlated with primary locomotion behaviors; moreover, network activity as a whole encodes behaviors better than the activity of individual neurons [13,14,47]. Many additional neurons and circuits have been linked to specific higher-order behaviors such as roaming and dwelling, local and global search, quiescence, chemotaxis, and the escape response [30,31,42,48,49]. Still, the question remains:

*How do these neurons work together to determine behavior, sustain different behaviors, and switch between them?*

Our data-driven, biology-based network model presents a dynamical picture different from those obtained by previous authors: our results do not fit with classical paradigms such as limit cycles, which are too simplistic given that forward-reversal transitions occur at irregular times. As our core system is driven by more than a hundred signal neurons with

diverse functionalities, our simulated premotor dynamics are complex and varied, with seemingly random switching statistics that match what is observed in imaging data. We have benefited from many extensively labeled whole-brain imaging datasets that became available only recently and that we used to fit model parameters [14]. These datasets have enabled us to build a model that is directly connected to anatomy and has more realistic model outputs.

A list of models related to ours is included at the end of the Introduction. There is little basis for comparison with the more abstract, phenomenological models. Below, we compare our results with a few that are more biophysical and/or data-driven.

The dynamical picture we present differs from those in Refs [17,23–25], the outputs of which do not match well whole-brain imaging activity in data; specifically, these models do not reproduce the stochastic switching phenomena. For example, the worm model in [24] swims forward; in [25], the model also swims forward, but toward an attractor, with no stochastic switching. The dynamics in [17,23] tend to periodic cycles that underlie forward locomotion in their models.

Ref [26] constructed a model from whole-brain imaging data that was collected while the worm was subjected to sensory stimuli in the form of periodic changes in NaCl concentration, which elicited a response in some sensory neurons. While locomotion neurons receive input from all of their presynaptic neighbors, Ref [26] found that the neural activity of the premotor neurons resided in separate motifs from the motifs capturing the sensory response to NaCl concentration changes. From this, they deduced that noise was necessary to produce realistic premotor activity. Our model leads to a different hypothesis: stochastic switching in our model occurs as a direct consequence of input from a (large) group of sensory and interneurons we call "signal neurons". Our findings are not necessarily in conflict, as the signal neurons that act as stimuli in our model are a more comprehensive set of stimuli than the neurons that responded to NaCl concentrations in Ref [26].

Finally, our model is less detailed, with fewer parameters, than, for example, Ref [25]. We have chosen to work with a parsimonious model that could be analyzed in a transparent manner, as our aim in this study is to discover mechanisms and to posit an explanation for how the premotor network integrates sensory information to produce realistic dynamics.

Because this model is semi-realistic, and it is analyzable, it has the potential to shed light on how connection type, structure, and signal input contribute to *C. elegans* locomotion. We find, for example, that gap junctions and synaptic connections support stochastic switching in different ways. By simulating the core neuron dynamics with and without activity from different signal neurons, we demonstrate that a small group of neurons is disproportionally responsible for inducing switches between forward and reversal locomotion, although the signals initiating switches are distributed among many neurons. Systematically ablating the core neurons reveals that not all ablations have an equal effect and that ablating the RIB and AIB neurons had the most significant impact on the dynamics. We observed also how different switching statistics results in qualitative differences in simulated locomotory paths.

## Challenges in building a mechanistic network that is data-driven, biophysical, and analyzable

One of the challenges in building such a model is the extraction of a suitable subnetwork. *C. elegans* neurons are highly recurrently connected; they do not process information in a feed-forward manner. Many premotor neurons are known to be "hubs" in the network with extensive connections [29]. These network features obfuscate which subset of neurons is responsible for determining locomotion—yet analyzability demands clarity and simplicity. An innovation in this paper is to select a core group of neurons of manageable size and model their dynamics as they receive input from a larger group of neurons (called signal neurons) the outputs of which we glean directly from data. We propose that approximating networked systems as a core system driven by external input can be a paradigm for modeling complex network dynamics in mathematical biology.

While it is far more realistic than studying a closed circuit driven by one or two input sources, the approach we have introduced is not without limitations: To keep the size of the core group manageable, we have had to omit a significant amount of feedback from the core neurons to the signal neurons. Because we view the premotor neurons as primarily

driving the motor neurons, we have neglected feedback from motor neurons. The exclusion of indirect signal transmission between core neurons via pathways that involve neurons outside of the core group is another major drawback, one that became all too apparent when comparing experimental and model signal propagation maps, as we have discussed earlier. We expect that model accuracy will improve if we fit the dynamics of a larger set of neurons (than the current core group), use more detailed models of individual neurons, and allow nonlinear interactions. This, however, will introduce more parameters, which in turn may require more data to constrain than what is currently available. See also the first paragraph under Ongoing and future work, where we discuss a specific way forward.

Our model is also constrained by data availability. We do not include neurons for which we lack whole-brain imaging data. Specifically, we exclude the PVC and DVA neurons despite their strong recurrent connections with the core set. Many additional neurons are rarely or never labeled in whole-brain imaging data which means that their contribution to the dynamics is not represented [14]. We expect that the accuracy of the model would improve if these neurons could be included.

A major challenge in modeling biological systems is integrating disparate and inconsistent experimental evidence (and/or inferences from other computational studies). An example of the uncertainty we have encountered is in the signs of synaptic weights in the *C. elegans* connectome. For example, Ref [34] contains an extensive set of experimentally determined polarity predictions using gene expression data. Though many of the polarities are labeled as complex due to the existence of both excitatory and inhibitory neurotransmitter-receptor pairs between the pre- and postsynaptic neurons, overall they predict more excitatory than inhibitory connections (Fig 10). In contrast, Ref [22] predicts, based on a computational model, that most interneurons in the circuit controlling locomotion (AVA, AVB, AVD, AVE, DVA, PVC) are inhibitory [21,22]. Not all experimental studies agree on sign predictions either. For example, the synaptic connection from RIM to AVB is excitatory according to Ref [34] but inhibitory according to Ref [42] (Fig 10). Additionally, there is not a clear relationship between functional connectivity and anatomical connections [50]. Because we cannot infer synaptic signs and weights from signal propagation data, as is available from Ref [39], we do not include the Fig 7 functional weights in the comparison of polarities.

Because different studies predict different synaptic polarities, it is unclear how to use previous studies to constrain model parameters; our predictions were obtained independently through regression based on whole-brain data. Nor is it possible—given these discrepancies—to match all existing data. Fig 10 shows synaptic signs and weights between core neurons in our model (grid box color) compared to the polarities predicted by several previous studies. As can be seen, our synaptic predictions match some previous predictions and deviate from others. For example, we predict inhibitory synapses from forward neurons AVB and RIB to reversal neurons AVE and AIB, and mostly excitatory synapses from the reversal neurons to other reversal neurons, matching fairly well the findings of Ref [34]. On the other hand, RIM to AVA is inhibitory in our model and Ref [16] but excitatory in Ref [34].

We remark that despite the discrepancies in polarity predictions, our model reproduces stochastic switching behavior, suggesting that such behavior is very robust.

## Summary and conclusion

- From a highly interconnected network such as the *C. elegans* central nervous system, it is *a priori* unclear if one can meaningfully isolate a small subnetwork for detailed study. This challenge is especially severe in cases such as this, when the subsystem that we wish to model is intermediary between inputs and outputs. We propose that a suitable framework may be that of *driven dynamical systems*.
- In this paper we have focused on premotor neurons, modeling them as a small network of core neurons driven by a much larger collection of "signal neurons". Our model reproduces switching dynamics, which underlie *C. elegans* locomotion, far more realistically than previous dynamical models, and it correctly predicts the influence of many neurons, though the model was not built to reflect the behavior of any one neuron.

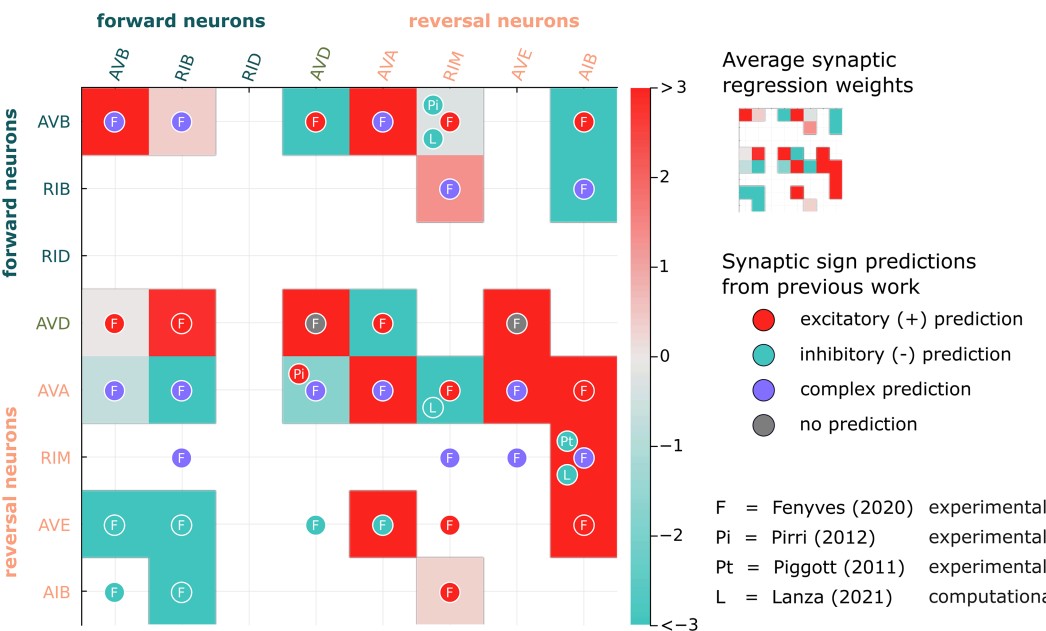

**Fig 10**. **Polarities obtained from regression compared to polarities determined from previous work [16,34,38,42].** Synaptic weights are grouped and averaged over each neuron class (e.g. AVBL and AVBR and combined to form AVB). Ref [34] determines polarities for synapses that are not in the Ref [12] dataset which is used to establish the location of synapses for the model in this study.

- Because the model is analyzable, we have used it to answer theoretical questions, such as the roles of gap junctions versus synapses. Model results suggest that, in the real worm, much of the seemingly random switching is in response to an array of inputs distributed across a large number of sensory and interneurons, some more influential than others. We hypothesize also that reverse motion is not antithesis to forward motion, rather it may serve as a brake to forward motion.
- Our model is not without limitations, however: a comparison with experimental functional connectivity maps revealed that more subtle responses in the network are missed. We conclude that overall, our modeling approach has yielded a useful though far-from-perfect approximation of the *C. elegans* premotor dynamics.

## Ongoing and future work

*On the technical side:* Although data availability is a limitation, using advanced statistical methods with the existing data may result in more accurate and reliable model fits. The identifiability analysis literature includes methods for fitting model parameters while also proving that the model fit is unique and accurate [51–55]. Identifiability has typically been achievable for smaller systems with well-defined terms and highly accessible measurements; it has not been generally established for larger nonlinear systems with few restrictions on terms. A useful future direction would be to explore the extent to which identifiability is achievable for neuronal systems given that the graph structure and known neuronal mechanisms will significantly constrain model possibilities.

*Towards advancing biology:* While our model captures transitions between the two primary locomotory states, forward and reversal, there are several other locomotory states whose neural correlates have been identified, such as dorsal and ventral turns, omega turns, and quiescence [13,15,56]. The methods developed in this paper can be used to study these

states. Our model can also be used to examine the impact of the environment on premotor activity through sensory neurons. Signals from the environment such as oxygen levels and chemical gradients are critical for chemotaxis and escape responses.

A biologically semi-realistic model such as ours can shed light on *dynamic mechanisms*, including mechanisms for sustaining and switching between different locomotion states, and for higher-order behavioral states such as dwelling and roaming. It can also be used to analyze the effects of ablations and identify the function of signal neurons with respect to locomotion.

Going forward, the driven dynamical systems paradigm proposed in this paper has many potential applications. Closer to this work is the modeling and analysis of other circuits in the *C. elegans* nervous system, and the integration of circuits for multiple functions into more comprehensive models. Further afield, the techniques developed here can be exported to other model organisms with available connectomic or large-scale imaging data.

## Methods

In the Results section, we pose a general model for *C. elegans* neural dynamics constructed using the connectome and experimentally observed intrinsic dynamics, and then fit model parameters using whole-brain imaging data. Here we fill in details omitted in the main text, mainly on (a) the modeling of intrinsic dynamics of individual neurons, and (b) how some of the parameters are fitted. We also provide more details on (c) network simulations such as those in Fig 3, and (d) our construction of locomotion paths in Fig 9.

### Model dynamics and parameters

As stated in the Results section, each of the core neurons, $\vec{\mathbf{x}} = [x_1, x_2, ..., x_n]^T \in \mathbb{R}^n$, has the following approximate dynamics for its calcium imaging brightness $x_i$ (neuron GCamp (z-scored)):

$$\tau \frac{dx_i}{dt} = f_i(x_i) + \beta \sum_{j=1}^{n+m} \mathbf{W}_{ij}(x_j - x_i) + \sum_{j=1}^{n+m} \mathbf{A}_{ij}\sigma(x_j) \qquad (2)$$

where $f(x_i)$ is neuron $x_i$'s intrinsic dynamics, $\beta \sum_{j=1}^{n+m} \mathbf{W}_{ij}(x_j - x_i)$ represents the influence from neurons that form gap junctions with neuron $x_i$, and $\sum_{j=1}^{n+m} \mathbf{A}_{ij}\sigma(x_j)$ represents the influence from neurons that are presynaptic to neuron $x_i$. $\sigma() =$ ReLU() is the ReLU activation function and $\tau$ is a global timescale variable. The dynamics of the signal neurons are not modeled, their dynamics are extracted from the whole-brain imaging time series.

**Intrinsic dynamics.** Voltage and current clamp experiments show that *C. elegans* neurons have a graded, nonlinear response to input current [32,33]. The degree of nonlinearity varies from neuron to neuron; some neurons smoothly depolarize or hyperpolarize from the resting membrane potential while other neurons are bistable [57]. Most *C. elegans* neurons do not exhibit action potentials as they lack voltage-gated sodium channels; however, a few neurons, such as the AWA class, are able to spike via voltage-gated calcium channels [57]. Due to voltage-dependent ion channels, *C. elegans* neurons tend to be highly sensitive to input around their resting membrane potential and become less sensitive to input far from their resting membrane potential (Fig 11A). Suppose that the dynamics for a neuron's voltage is $\frac{dV}{dt} = f(V)$. An approximation for $f(V)$ can be attained through voltage clamp experiments. The dynamics for a neuron's voltage when an input current is applied to the neuron is $\frac{dV}{dt} = f(V) + I$. Voltage clamps stabilize a neuron's voltage at a given level $V$, essentially setting the derivative equal to zero, $\frac{dV}{dt} = 0$, by applying a current $I$ that will satisfy this objective. The amount of current $I$ necessary to stabilize the neuron at a range of voltage levels $V$ provides an approximation for $-f(V)$. We approximate $-f(V)$ for the inter/motor neuron RMD with the cubic function $-f(V) = \frac{1}{8000}(V+70)(V+60)(V+50)$ (Fig 11B). This cubic function approximately matches the RMD steady-state I-V relation derived in Ref [33] (Fig 11A). The approximate function for intrinsic voltage dynamics, $f(V) = -\frac{1}{8000}(V + 70)(V + 60)(V + 50)$, indicates that there are two stable fixed points in

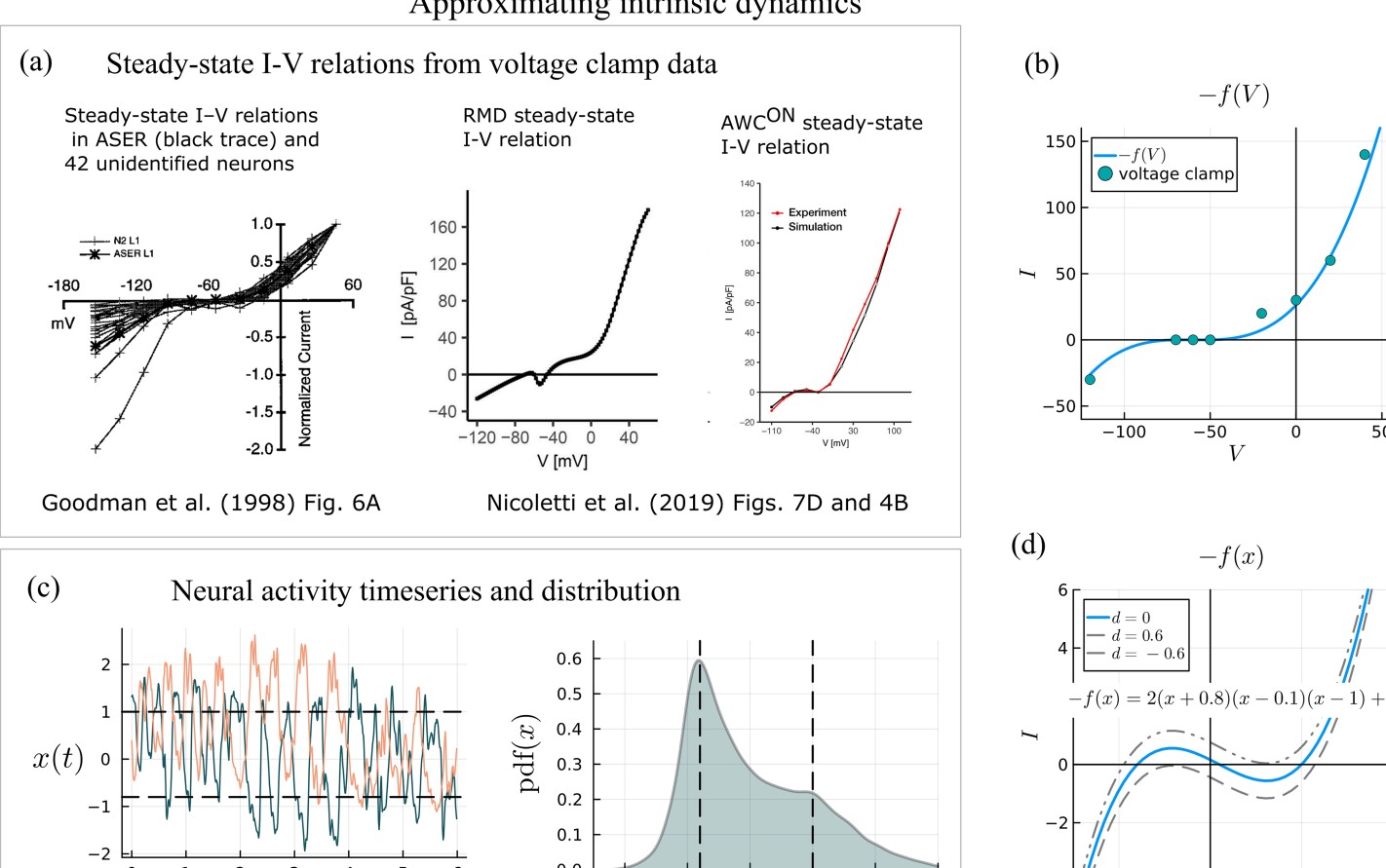

**Fig 11. (a) Voltage clamp experiments from Ref [32].** Reprinted from *Neuron*, 20(4), Goodman M, Hall D, Avery L, and Lockery S, *Active currents regulate sensitivity and dynamic range in C. elegans neurons*, 763–772, Copyright (1998), with permission from Elsevier. Voltage clamp experiments from Ref [33]. (b) Intrinsic dynamics approximated from voltage clamp data in terms of voltage, *V*. (c) Calcium imaging time series and distribution over premotor neurons. The pdf of the time series of the premotor neurons is used to set fixed points for the intrinsic dynamics, $x_{fp} = -0.8, 0.1, 1$. (d) Negative of approximate intrinsic dynamics, $-f(x)$, in terms of the calcium imaging variable, *x*.

the voltage dynamics at approximately $V = -70$ and $V = -50$ and an unstable fixed point at $V = -60$. This bistability is corroborated by experiments that show that RMD neurons have a plateau potential at approximately –35 mV in addition to its resting membrane potential at –70 mV [58].

While voltage clamp experiments can provide approximate forms for the intrinsic dynamics in terms of voltage, we need formulas for the intrinsic dynamics in terms of calcium imaging brightness, which poses a challenge as there is no function that directly transforms voltage to GCaMP brightness. We posit a nonlinear, cubic form for the intrinsic dynamics of each neuron in terms of the state variables for our model (GCaMP z-scores) using the following justification. Experiments measuring voltage levels and GCaMP (calcium imaging) concurrently in AWA neurons found that GCaMP fluorescence increased with rising voltage levels and decreased with drops in voltage levels, with a response that was smooth and delayed [57]. We estimate the average resting membrane potential and sensitivity range in *x*, the neuron GCaMP variable, by taking the distribution of *x* over the premotor neurons and finding peaks in the distribution (Fig 11C). We find the

most common value across neurons is $x = -0.8$, and so set this as the average resting membrane potential (Fig 11C). There is another bump in the distribution around $x = 1$, and so we set this to be the plateau potential stable state in the intrinsic dynamics with an unstable fixed point between the two stable fixed points at $x = 0.1$ (Fig 11D). The resulting cubic $f(x) = -2(x + 0.8)(x - 0.1)(x - 1)$ is a qualitative estimate for the intrinsic dynamics of interneurons in general. The locations of the fixed points and their stabilities vary from neuron to neuron—this is accounted for by introducing an individual bias term $\mathbf{d}_i$ for each neuron. Voltage clamps show that while the intrinsic dynamics of neurons have the same qualitative shape, the fixed point locations differ [32,33] (Fig 11A). We include the bias term $\mathbf{d}_i$ in the intrinsic dynamics to account for this variability (Fig 11D).

We set the intrinsic dynamics of each neuron in terms of $x_i$, without input from other neurons to be

$$\frac{dx_i}{dt} = f_i(x_i) = -2(x_i + 0.8)(x_i - 0.1)(x_i - 1) + \mathbf{d}_i \qquad (3)$$

where $\mathbf{d}_i$ varies from neuron to neuron. When $\mathbf{d}_i = 0$, the intrinsic dynamics have two stable states at $x_i = -0.8$ and $x_i = 1$ and one unstable state at $x_i = 0.1$. Allowing a different bias term $\mathbf{d}_i$ for each neuron allows the locations and stability of the fixed points to vary across neurons while keeping the same general form. Depending on the value of $\mathbf{d}_i$, one of the stable fixed points can become semi-stable or eliminated (Fig 11D).

The cubic form we use to approximate the intrinsic dynamics captures several key properties. Firstly, it allows for voltage bistability in interneurons. Plateau potentials allow neurons to have the same computational properties as Schmitt triggers [58,59]. Neurons that lack two stable fixed points due to the bias term still have richer computational properties than if they possessed linear intrinsic dynamics. Second, the cubic form acts to provide bounds on the voltage range of each neuron. Neurons do not have an unlimited capacity for hyperpolarization or depolarization; negative feedback pressures keep the voltage within a bounded range and our model captures this constraint with the negative cubic term $-x^3$. Saturation effects must be included to ensure the model produces realistic neural activity. Thirdly, the cubic form reflects the neurons' increased sensitivity to input near its resting membrane potential and plateau potential. The neurons' sensitivity to input varies in a voltage-dependent manner which is consistent with experimental data [32].

The cubic form we use for the intrinsic dynamics of the *C. elegans* neurons has similarities to the voltage dynamics in models for spiking neurons. Both the FitzHugh–Nagumo and Hindmarsh–Rose models of neuron dynamics approximate the voltage dynamics with a cubic function [60–62]. One of the key differences between these models and our model is that models of spiking neurons incorporate recovery variables that capture spiking and the instability of the high-voltage state. *C. elegans* neurons do not spike and can be stable at a high-voltage state so we approximate each neuron's intrinsic dynamics without using recovery variables [58,59].

**Gap junctions and synaptic weights.** The second term in Eq 2, $\beta \sum_{j=1}^{n+m} \mathbf{W}_{ij}(x_j - x_i)$, captures the influence of neurons connected to neuron $x_i$ via gap junctions. $\mathbf{W}$ is a symmetric matrix that holds the relative strengths of gap junctions between pairs of neurons; these weights are taken from the connectome [12]. The weights in $\mathbf{W}$ are applied to the difference between $x_j$ and $x_i$, meaning that the activity of neurons connected via gap junctions are pressured to equalize. The entire term is scaled by $\beta$ which determines the relative contribution of gap junctions to the overall dynamics; we fit $\beta$ with a parameter sweep over many fits with whole-brain imaging data.

The third term in Eq 2, $\sum_{j=1}^{n+m} \mathbf{A}_{ij}\sigma(x_j)$ captures the input from neurons presynaptic to neuron $x_i$. $\mathbf{A}$ is a nonsymmetric matrix that contains the signed, weighted, and directed synaptic inputs between pairs of neurons. The edge graph for synaptic connections in *C. elegans* is derived from [12].

**Parameter fit.** We determine $\beta$, $\mathbf{A}$, and $\mathbf{d}$ simultaneously by performing multiple linear regressions to approximate $\mathbf{A}$ and $\mathbf{d}$ for different values of $\beta$ across 22 datasets selected from Ref [14]. Our criteria for the selection of these 22 datasets is that the dataset must contain at least one of the following forward core neurons—AVBL, AVBR, RIBL, or RIBR—because they are sparsely labeled in the datasets and one neuron in each class of the reversal neurons and AVD. To

reduce the error from missing neurons we perform data replacement on missing time series when possible by substituting missing time series with the time series of highly correlated neurons. Certain pairs of left/right neurons are consistently highly correlated and therefore one neuron's time series can be used as a proxy for the other if one neuron in the pair is missing. We label neuron pairs as highly correlated if they are on average at least 70% correlated in the datasets where they both appear; 30 neurons fit this criterion. For these highly correlated neurons, when one of the neurons is missing we use the time series of the other neuron as a proxy in the regression.

We set $\tau = 1$ and fit the remaining parameters. From Eq 2 we subtract the default intrinsic dynamics and gap junction terms from the derivatives, leaving the synaptic weights and bias terms on the right-hand side,

$$\frac{dx_i}{dt} + 2(x_i + 0.8)(x_i - 0.1)(x_i - 1) - \beta \sum_{j=1}^{n+m} \mathbf{W}_{ij}(x_j - x_i) = \sum_{j=1}^{n+m} \mathbf{A}_{ij}\sigma(x_j) + \mathbf{d}_i. \tag{4}$$

Eq 4 provides us with a reformulation of Eq 2 where all parameters on the left-hand side (LHS) are determined with the exception of $\beta$. We fix different values for $\beta$ and then perform linear regressions for the 22 datasets to approximate $\mathbf{A}$ and $\mathbf{d}$ using snapshots from the calcium imaging time series of all neurons $\mathbf{x}(t)$ acquired from Ref [14].

We perform a parameter sweep for $\beta$ and select the $\beta$ value that minimizes a combination of the regression error and the reconstructed derivative error across all datasets and across core neurons (Fig 12A). The regression error, error$_{regress}$, is the L2 norm of the difference between the timseries of the left-hand side and right-hand side of Eq 4. The reconstructed derivative error, error$_{deriv}$, is the norm of the difference between the time series of the derivative and the reconstruction of the derivative, Eq 2. The hybrid error is a combination of the regression error and the derivative error, error$_{hybrid}$ = error$_{regress}$ + $0.05 *$ error$_{deriv}$. The average regression error, derivative error, and hybrid error are shown for varying $\beta$ values

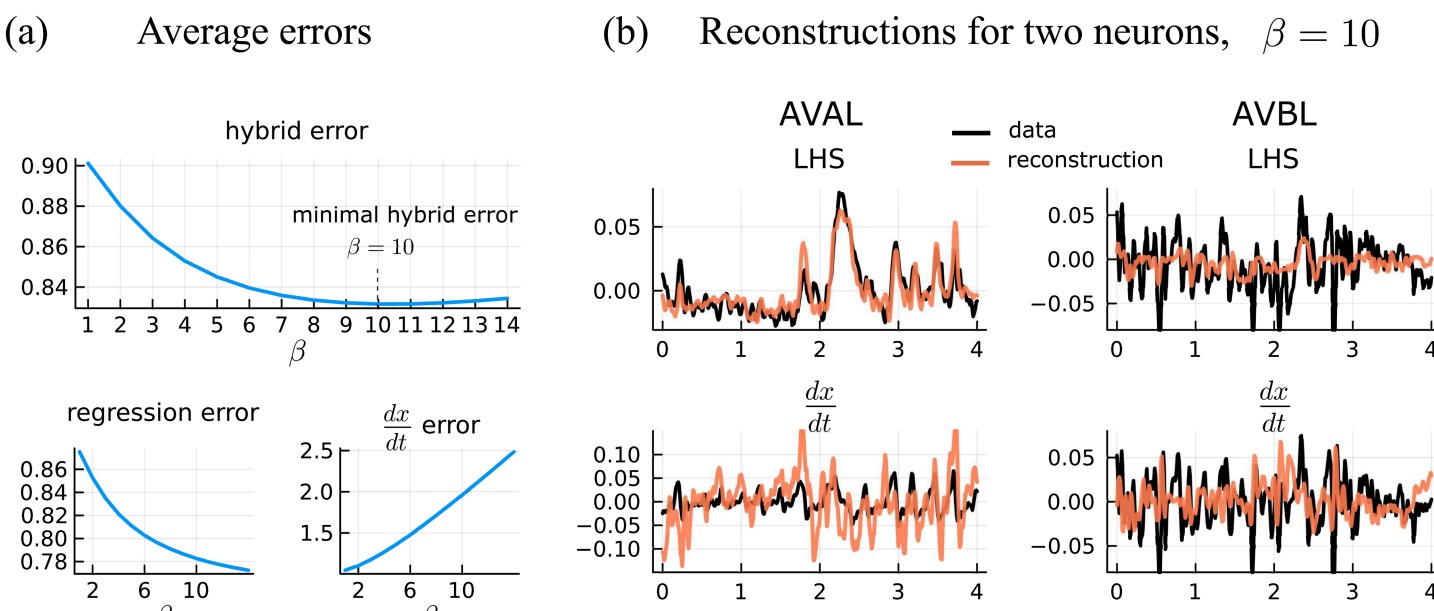

**Fig 12**. **(a) Average hybrid error as a function of $\beta$.** The average error is a combination of the regression error and the derivative reconstruction error. Derivative and reconstruction errors are the L2 norm of the difference between the LHS and RHS of Eqs 2 and 4 respectively. (b) Example regression LHS and derivative reconstruction for AVAL and AVBL for $\beta = 10$.

in Fig 12A. As an example of error for individual neurons, the top row of Fig 12B shows the time series data of the left-hand side of Eq 4 with its reconstruction for neurons AVAL and AVBL. The bottom row shows the time series data of the derivatives of AVAL and AVBL along with the derivative reconstructions, Eq 2.

We observe the hybrid error across datasets and core neurons is minimized at $\beta = 10$ (Fig 12). We set $\beta = 10$ and then perform linear regressions for $\mathbf{A}$ and $\mathbf{d}$ estimates across the 22 datasets. We use the average of the weights and biases across all datasets as the synaptic weights and biases in the general model.

**Whole-brain imaging datasets.** We use calcium imaging time series data from Ref [14]. Each dataset consists of calcium imaging time series data for 64–106 labeled neurons [14]. Access to this data is provided through the Worm Wide Web database: https://wormwideweb.org/dataset.html.

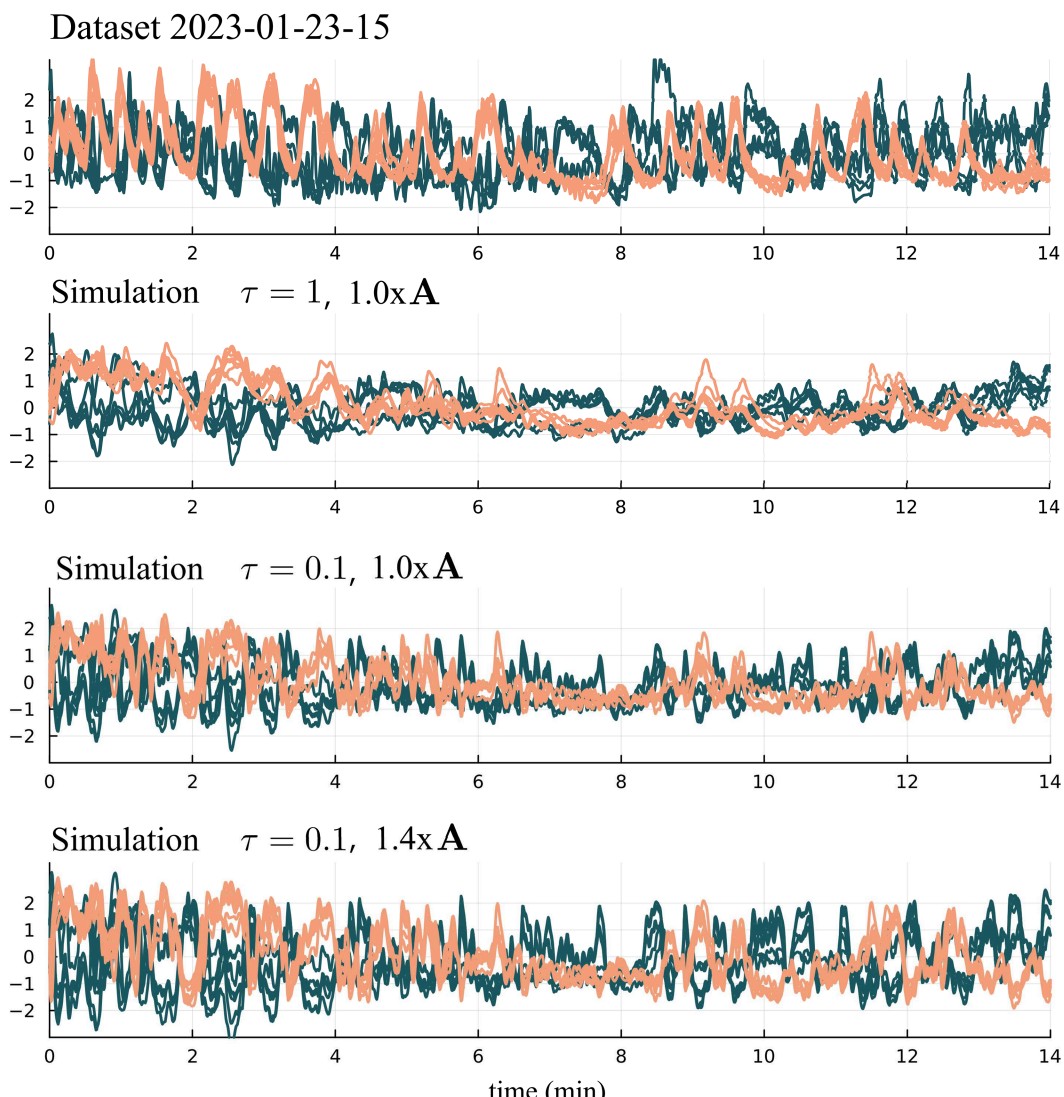

**Fig 13**. **Dataset compared to simulations with different timescales and magnification factors.** Decreasing the timescale parameter $\tau$ makes the system respond quicker to stimuli. Because the system is receiving partial signals we also magnify $\mathbf{A}$ by 1.4x. Increasing the amount of stimulus to the premotor neurons increases their average activity levels.

**Premotor network simulation.** We simulate the dynamics of core neurons by integrating Eq 2, using the time series of signal neurons as input. We perform numerical integration in Julia using the Runge-Kutta-Fehlberg (RKF45) method. We fit the timescale parameter by finding $\tau$ such that the simulated neural dynamics have the same reaction times as those observed in the neural recordings across the datasets shown in Fig 14. The closest reaction time results from $\tau = 0.2$. Because each dataset contains traces for only a subset of neurons, each simulation is performed with input from a subset of the theoretical input. To help compensate for the missing input, we amplify the existing synaptic input by a magnification factor. We simulate the dynamics using increasingly large magnification factors and find a magnification factor of 1.4x brings the core neurons to the correct activity level.

An example of how $\tau$ and the magnification factor affect the simulation is shown in Fig 13. Neural recordings from premotor neurons in dataset 2023-01-23-15 are compared to simulations performed with varying timescale parameters $\tau$ and magnification factors for the synaptic input (Fig 13). Decreasing the timescale parameter $\tau$ makes the system react quicker to input. Increasing the magnification factor increases the amount of input applied to the neurons which results in higher average activity levels in the neurons. This phenomena is similarly observed in the other datasets. By decreasing the timescale parameter as well as magnifying the input from **A** we get simulated behavior that more closely matches the premotor neural recordings.

Using a single set of parameters, the general model reproduces the core neural activity observed in many datasets (Fig 14). While the parameter values remain the same, the input provided to the core neurons varies from dataset to dataset, resulting in different network activity.

## Locomotion path simulation

We simulate example locomotion paths for the forward/reversal behavior sequences, obtained from the data, using a simplified version of the procedure outlined in Ref [41]. Three sample paths for each dataset are initialized at the red marks. The locomotion cycle consists of a forward run, a reversal, a turn following the reversal, and then the resumption of forward motion. The forward speed is set to 0.15 mm/sec and the reversal speed to 0.075 mm/sec. The probability of a regular turn is 65% while the probability of an omega turn is 35%. Regular turn angles are selected from a uniform distribution ranging from $-90°$ to $90°$. Omega turns are deeper turns; they are selected from a uniform distribution ranging from $90°$ to $270°$. Because turn angles are random, the three simulated paths for each dataset differ despite having the same forward/reversal sequences and durations.

## Acknowledgments

The majority of this work was done while MM was an NSF Mathematical Sciences Postdoctoral Research Fellow at the Courant Institute of Mathematical Sciences at New York University.

## Additional simulations

Fig 14 shows the core neuron activity from six different datasets compared to simulated core neuron activity when receiving signal input from these datasets. An experimental heat shock was applied at the 8 minute mark in datasets a-e, which has the effect of increasing the probability of entering the reversal state after the heat shock (Fig 8) [14].

## Dataset 2023-01-23-15 promoter and suppressor neurons

Applying the same process as in Fig 5, we ranked the signal neurons in dataset 2023-01-23-15 [14] by the extent to which they excite or inhibit the forward or reversal neurons through synaptic connections and then systematically set the top promotors and suppressors to zero during the simulations (Fig 15). Similar to the results for dataset 2023-01-05-01 (Fig 5), the simulations using dataset 2023-01-23-15 indicate that the switch from reversal to forward is instigated primarily by signal neurons promoting the forward core neurons rather than inhibiting the reversal core neurons (Fig 15A–15B).

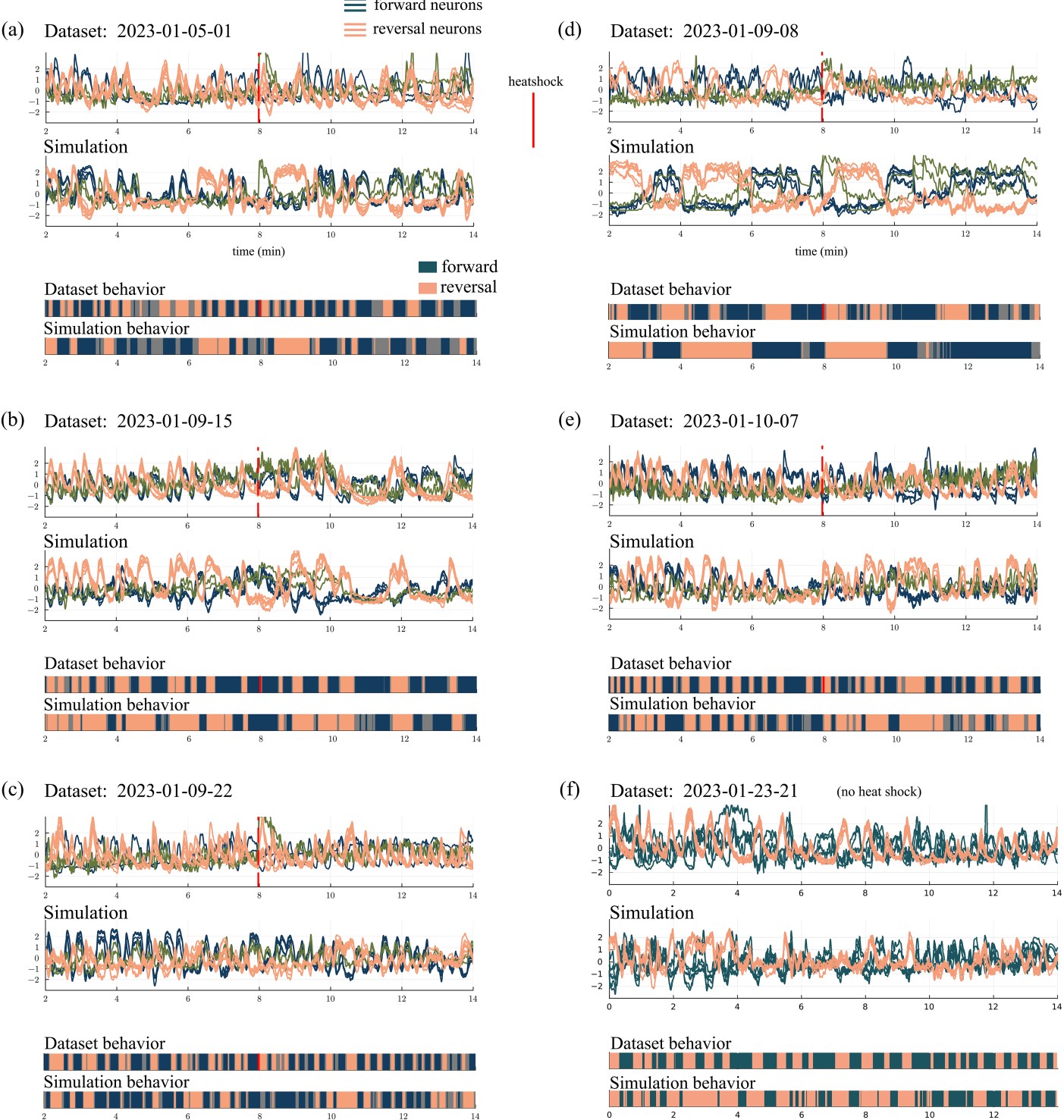

**Fig 14**. **Time series of core neurons in whole-brain imaging data compared to the simulation using signal neuron input.** Behavior is measured as the dominant cluster (forward or reversal). $\tau = 0.2$, magnification on synaptic input is 1.4.

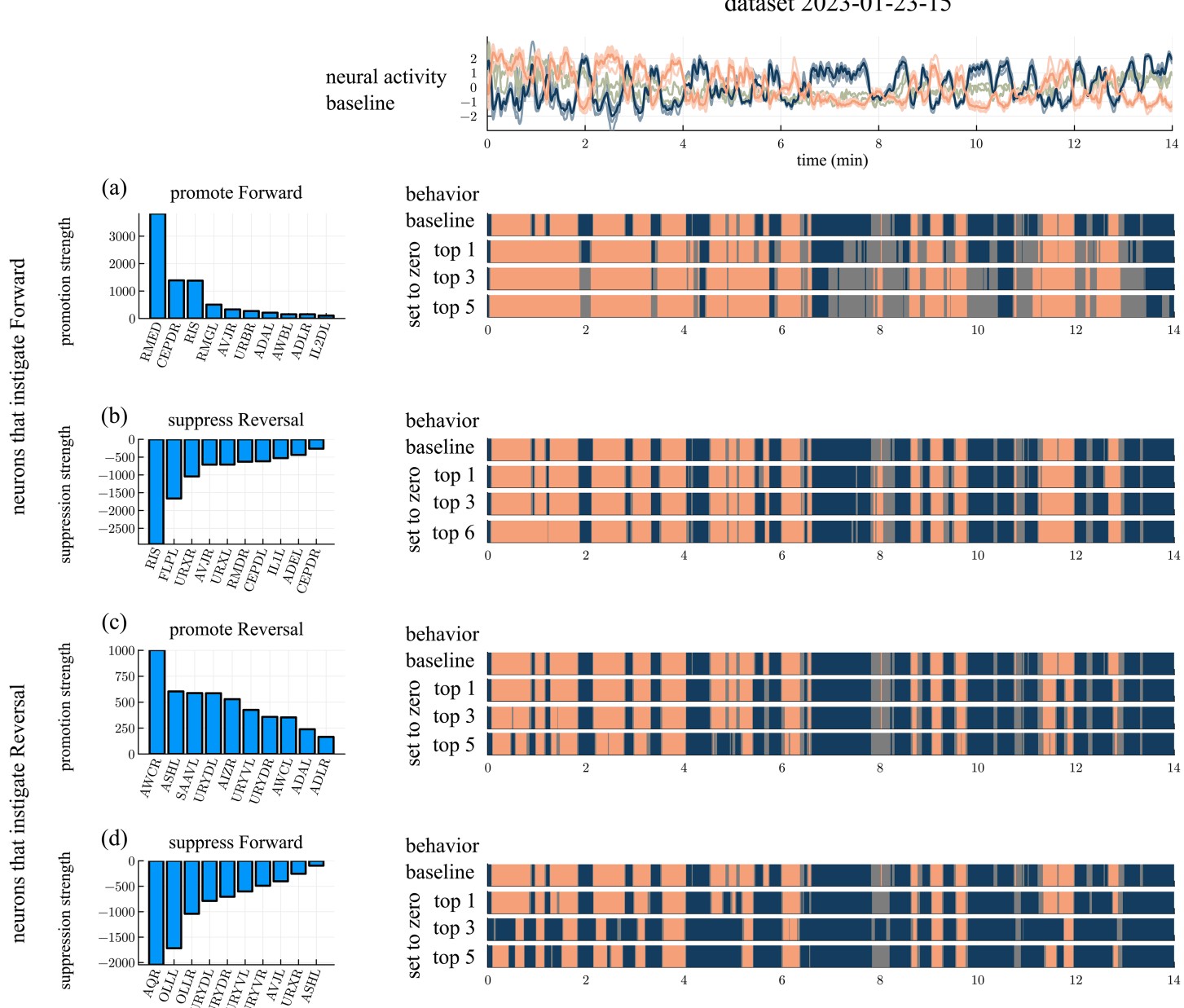

**Fig 15**. **Simulations using signal neurons from dataset 2023-01-23-15 [14].** Promoter and suppressor neurons from dataset 2023-01-23-15 are ranked by the strength of their promotion or suppression. (a) Signal neurons that promote the forward neurons. Behavioral time series with the top promoters set to zero. (b) Signal neurons that suppress the reversal neurons. Behavior with the top suppressors set to zero. (c) Signal neurons that promote reversal neurons and resulting behavior with the top reversal promoters set to zero. (d) Signal neurons that suppress the forward neurons and resulting behavior with the top forward suppressors set to zero.

The switch from forward to reversal appears to be more strongly instigated by forward suppressors rather than reversal promoters for dataset 2023-01-23-15 (Fig 15C–15D). The switch signal is distributed across more neurons for dataset 2023-01-23-15—neutralizing more promoters and suppressors is necessary to reduce the duration and frequency of reversals.

## Inferring behavior from core neuron activity

Locomotion state (behavior) is inferred from simulated neural activity by determining which core neuron cluster, forward or reversal, has a higher activity level. We average the five forward neurons and eight reversal neurons in each simulation and dataset to get low-dimensional variables $F(t)$ and $R(t)$ which describe the activity of each cluster. The average forward neuron activity is $F(t) = \frac{1}{N_F} \sum_{i=1}^{N_F} x_{Fi}(t)$, where $x_{Fi}(t)$ are the time series of the forward neurons and $N_F = 5$. The average reversal neuron activity is $R(t) = \frac{1}{N_R} \sum_{i=1}^{N_R} x_{Ri}(t)$, where $x_{Ri}(t)$ are the time series of the reversal neurons and $N_R = 8$. The difference of the cluster averages, $z(t) = F(t) - R(t)$, is a single, low-dimensional variable that is highly correlated with the true velocity of the *C. elegans* (Fig 16B, 16D). We use the difference of the cluster averages as a proxy for velocity in the simulations. If the difference in the cluster activity is small, $-0.5 < z < 0.5$, the corresponding behavior is labeled as being in the pause state, while if the difference is larger the behavior is labeled as either forward or reversal.

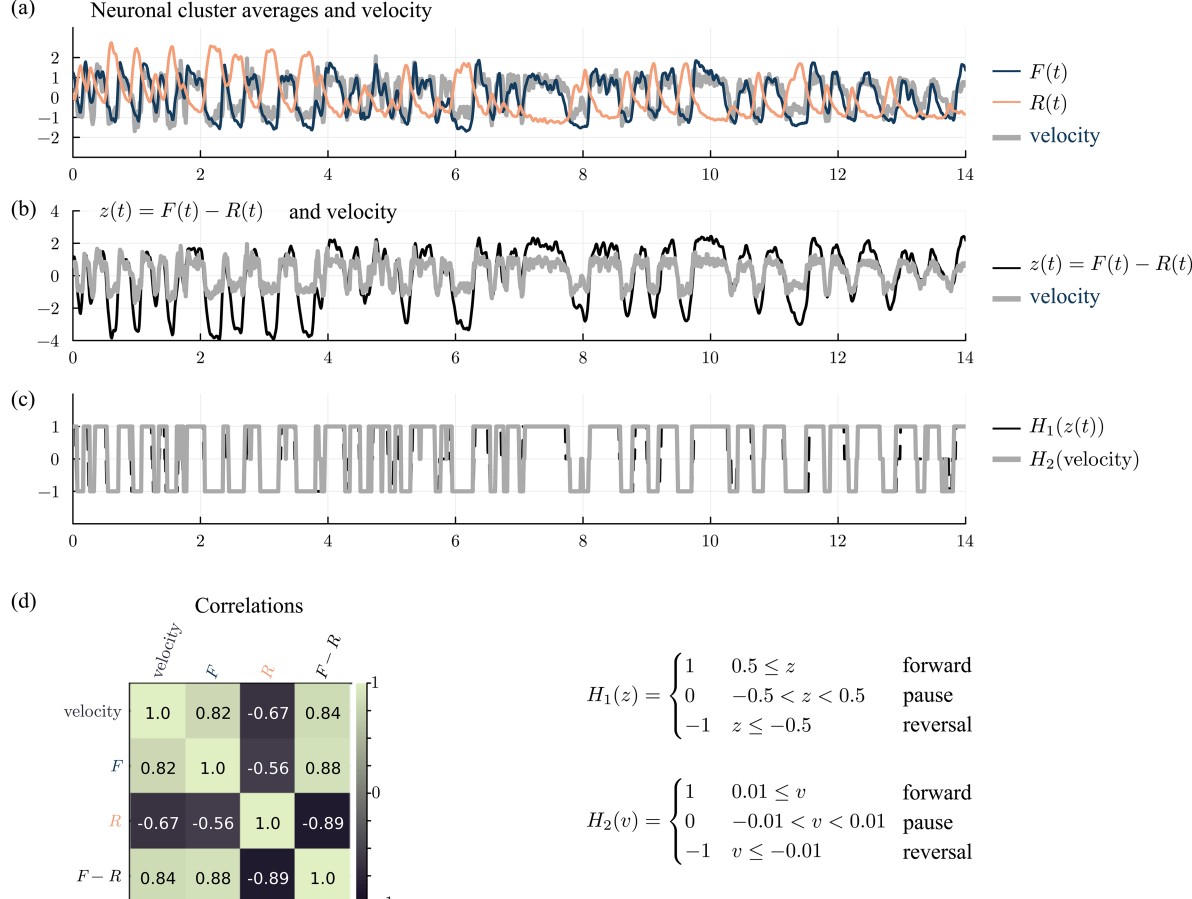

**Fig 16. (a) Forward and reversal neuron average activity levels along with velocity.** (b) Difference of cluster averages and velocity. (c) Behavioral state classification of the difference variable ($z(t)$) and true velocity. (d) Correlations between the true velocity and measures taken from the neural activity—$F(t)$, $R(t)$, and $F(t)$–$R(t)$.

## Author contributions

**Conceptualization:** Megan Morrison, Lai-Sang Young.

**Data curation:** Megan Morrison.

**Formal analysis:** Megan Morrison.

**Funding acquisition:** Megan Morrison, Lai-Sang Young.

**Investigation:** Megan Morrison.

**Methodology:** Megan Morrison, Lai-Sang Young.

**Software:** Megan Morrison.

**Supervision:** Lai-Sang Young.

**Validation:** Megan Morrison.

**Visualization:** Megan Morrison.

**Writing – original draft:** Megan Morrison, Lai-Sang Young.

**Writing – review & editing:** Megan Morrison, Lai-Sang Young.

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
