## [Decision Letter · Decision Letter 0]

1 May 2025

PCOMPBIOL-D-25-00009

A data-driven biophysical network model reproduces *C. elegans*  premotor neural dynamics

PLOS Computational Biology

Dear Dr. Morrison,

Thank you for submitting your manuscript to PLOS Computational Biology. We apologise for the delay in review of your manuscript, due to waiting for a fully complete review by one invited reviewer. Consequently, I have personally reviewed the manuscript to allow a decision to be made - my comments are included below as Reviewer #3. Should the still outstanding review be completed, we will forward this to you as soon as possible.

After careful consideration, we feel that your manuscript has merit but does not fully meet PLOS Computational Biology's publication criteria as it currently stands. Therefore, we invite you to submit a revised version of the manuscript that addresses the points raised during the review process.

Please submit your revised manuscript within 60 days Jul 01 2025 11:59PM. If you will need more time than this to complete your revisions, please reply to this message or contact the journal office at ploscompbiol@plos.org. Please include the following items when submitting your revised manuscript:

We look forward to receiving your revised manuscript.

Kind regards,

Barbara Webb

Academic Editor

PLOS Computational Biology

Hugues Berry

Section Editor

PLOS Computational Biology

**Journal Requirements:**

At this stage, the following Authors/Authors require contributions: Megan Jean Morrison, and Lai-Sang Young. Please ensure that the full contributions of each author are acknowledged in the "Add/Edit/Remove Authors" section of our submission form.

4) Please ensure that the funders and grant numbers match between the Financial Disclosure field and the Funding Information tab in your submission form. Note that the funders must be provided in the same order in both places as well.

**Reviewers' comments:**

Reviewer's Responses to Questions

**Comments to the Authors:**

Reviewer #1: Morrison et al reports an ODE model that captures the dynamics of a main subset of the command interneurons in C. elegans. The model is simple and easy to understand, and it reproduces actual neural activity well. Mathematical modeling that follows actual data is very important and should be highly evaluated. Analyzing the model predicted the different role of chemical synapses and gap junctions on neural activities. However, the paper seems to require additional evidence to demonstrate the importance of the model.

Major:

1) The proposed model covers a major subset of the command interneurons, but other neuronal activities are treated as inputs to the model. If these inputs include clear signals that indicate forward or backward movement, it would be easy to switch between forward and backward movement even without the proposed model. In fact, the experimental data used by the authors (Atanas et al 2023) includes experiments in which aversive heat stimulation was applied to induce strong backward movement. It is necessary to examine the properties of the input signal in addition to the presence or absence of such thermal stimulation. For example, as in Figure 7 of Hallinen 2021 (doi: 10.7554/eLife.66135), it may be possible to examine how the neural activity in the model is related to the main components extracted by principal component analysis of the input signal. By separating whether the transitions between the forward and backward states in the model are strongly dependent on the input signal, it becomes easier to understand how the state transitions occur, and it seems to be a good example that shows the importance of the proposed model.

2) The authors claim that "The dynamics produced are complex: they are neither completely random as in the Markov chains models nor are they limit cycles as in earlier dynamical systems models." However, the dynamics of the proposed model are heavily dependent on external forces, so a superficial comparison is not valid. It is necessary to analyze and compare how state transitions occur in the model, independently of the model's input. I would like to see how the state transitions occur in the proposed model, after extracting the time periods when the effect of external forces is weak, as shown in comment 1). In addition, the differences in the roles of individual neurons are also interesting, but they have not been analyzed at all. For example, it is thought that RIM stabilizes the forward and backward states (Sordillo A., et al., 2021, eLife). Also, is AVD, which the authors incorporated into the model, important for switching between forward and backward? To investigate the role of these neurons, it may be useful to perform an ablation study of RIM and AVD in the model. I believe that such additional analysis is essential for the authors' main question, “How do these neurons work together to determine behavior, sustain different behaviors, and switch between them?”

3) The time series used in Figure 5 is the actual data taken from Atanas et al., and it is unclear how the ODE model proposed by the authors contributes to this analysis. In addition, it is known that the duration of forward movement roughly determines whether the animal is dwelling or roaming, so it is necessary to show whether the proposed model can reproduce the distribution of the duration of forward movement.

4) Regarding to Figure 6: The weight from ASEL/R to the downstream circuit is just not visible. In the experimental dataset, there is no obvious input to ASEs, so it would be difficult to correctly estimate the weight to AIB. The apparent weight from ASER to AIB is already report (Sato H., et al., 2021, 10.1016/j.celrep.2021.109177).

Overall, it would be difficult to analyze responses to stimuli based on Atanas' dataset. It seems that a detailed analysis of the heat avoidance experiment conducted by Atanas et al. would be possible, but it is not mentioned at all in this paper. AWC is activated very transiently by thermal stimulation (Figure 7 in Atanas 2023). However, since other thermosensitive neurons are also activated (or inhibited) at the same time by thermal stimulation, it seems difficult to correctly estimate the effect of AWC alone.

5) Regarding to Figure 7: If you want to predict synaptic polarity, you need to refer to the actual measurement values (Randi F., et al., 2023, Nature). The references listed are only guesses and are weak as evidence. The fact that there seems to be no relationship between the number of synapses and the weight of synaptic connections is alrady mentioned in several papers including the NeuroPAL paper (Yemini E., et al., 2021, Cell). Also, since there is a strong correlation between neural activity, it seems necessary to consider multicollinearity for parameter estimation. If you simply estimated the parameters from a small number of real data without taking some measures, such as dimensionality reduction, the estimated values will be inaccurate. You might also want to check the sloppiness (identifiability) of the parameters.

Minor

6) Text in the legend of figure 1 contains more than legend.

7) In figure 3b, the authors estimate the behavioral labels based on the difference between the average neural activity F(t) and R(t). This method is need to be validated. Since the behavioral labels of the real data can be estimated using the same method, it is necessary to show whether they match the real data (velocity in the dataset).

8) In figure 3d, the distribution of RID appears to be different between the real data and the model. Since the distribution across all time is affected by the ratio of different behaviors in each trial, it may be necessary to show a stratified histogram for each behavior label and discuss the consistency.

9) Regarding Figure 5: dwelling and roaming are things that switch over time within the same individual, and they change in about 10 minutes (Flavell S.W., et al., 2013, Cell, doi: 10.1016/j.cell.2013.08.001). The data is obtained from Atanas 2023, but are these really dwelling and roaming? It is possible that you are simply looking at individual differences.

10) Figure 11 may be more appropriate as a supplementary figure.

11) Since there are comprehensive neural and behavioral models such as MetaWorm (currently BAAIworm, https://doi.org/10.1038/s43588-024-00738-w), comparisons with these models may be informative.

12) The conversion between membrane potential and GCaMP values seems to be based on the assumption that the binary state transitions seen in GCaMP measurements are similar to the stationary values of the I-V curve. If there are any evidence that justify this assumption, it would be better to cite the literature.

Reviewer #3: This paper exploits the richness of C. elegans data to take a somewhat novel approach to modelling the neural circuits that underpin switches in behaviour. More specifically, it uses previous identification of premotor neurons, and connectome data, to establish a small circuit model (of core neurons) whose inputs are given by whole brain recordings of upstream neurons (signal neurons). The connectivity parameters in the circuit are fitted using the same neural data, and its dynamic behaviour evaluated.

The main limitation I see in this approach is that it is not clear what, if anything, specific to the core circuit actually accounts for the ability to produce similar dynamic behaviour. That is, it seems possible that a completely generic neural network could be tuned/trained to produce the appropriate (simple) output (a dynamically switching tendency towards forward or backward motion) given the very rich input information.

The presented results make a small start towards exploring this by examining the circuit performance with either gap junctions or synaptic connections left out, and this is the most interesting part of the paper. But a much more thorough examination is needed, e.g., to see what is the minimal circuit that can produce realistic results, to ablate individual neurons in the circuit (not just the inputs) and test the consequences, and to establish what happens as the parameter values are varied.

The model also seems to make a number of other serious compromises with respect to realism, such as no recurrency from core to signal neurons, eliminating some potentially relevant neurons due to lack of data, not modelling motor output or motor feedback, ignoring neuromodulation, etc. Of course, any practical model has to make compromises to be tractable, and some of these limitations are discussed appropriately, but they also limit what can actually be learnt from the model. For example, while it is interesting to see the predicted polarity for neural connections obtained from the data fitting approach, and to discuss how these align with or contradict previous accounts, it seems likely the fitted connection weights, and their polarity, could be very different if any of the above factors were included.

I also felt the paper overall did not give a sufficiently deep embedding of how this work relates to the extensive range of previous models of this system. Instead these tended to be described somewhat briefly and dismissively.

Abstract:

“minimally parameterized, biophysical dynamical systems mode” - I am not sure that I agree that the degree of tuning to data actually used to set the parameters here should be described as “minimally parameterized”. It is also not fully evident in what sense the model is “biophysical”. The only support is in the claim (in the methods) that the cubic approximation for the intrinsic dynamics of the core neurons is similar to the form of models (e.g. Fitzhugh-Nagumo and Hindmarsh-Rose) of neural voltage dynamics. I agree the approximation does capture appear to capture the key properties, but this is not the same as having a biophysical justification for the elements of the equation.

Introduction:

It seems unnecessary to characterize the current state of understanding so negatively, e.g. “The neural mechanisms responsible for producing different behaviors and inducing transitions are not well understood, even in the simplest of organisms”. “relatively little is known” “the mechanisms that govern their collective activity remain to be understood”. I would prefer the authors to provide a clear outline of what is known, what has been hypothesized, and how the problem has been approached to date. From the introduction, a naive reader would be unaware of the extensive range of previous models of this system, as these mostly are referenced only in the discussion.

Line 26 on, it is good in principle to set out your modelling criteria, but other criteria might be advanced, so why are these chosen?

Line 107 on, the explanation for selecting neurons seems somewhat vague. Was there an explicit criteria, e.g., neurons shown to produce locomotion when individually activated (“command neurons”)? Neurons one synapse away from motor neurons (“premotor neurons”)?

Equation 1, the summation terms seem to imply that the inputs are the (W_ij) weighted gap junction inputs from both core and signal neurons, and the (A_ij) weighted inputs from both core and signal neurons; this is also what Fig 2a and 2b seem to show. Yet the text on lines 147-150 describes these terms in the equation as the “influence from all core neurons”. This creates confusion as I read the following parts under the assumption that A_ij is fitted to the data for the core neurons only, but (I believe) in fact it is also fitted to the signal data. It seems this could be sufficient to explain the matches in switching dynamics and higher order statistics shown in the results, irrespective of the structure of the core circuit.

The differential effects of synapses vs. gap junctions seems potentially interesting. But could the correct dynamics be recovered if the parameters were refitted for the circuit without synapses/gap junctions?

Behaviour over longer time durations: I found this hard to follow, particularly how it relates to the model provided here. It seems hardly necessary to provide a model to support the inference that more frequent switching results in shorter runs, and hence less dispersal. This part of the work seems unnecessary to include in the paper.

Lines 359-361, rather than dismissing these mathematical models, it would be more interesting to discuss if the current model gives more support to one of these abstractions over another.

Lline 411, can you comment more on how the accuracy of the model could improve. E.g. In what way would you test for a better fit to the data than that shown in fig 3 for the current model?

For clarity, this review was completed by the editor for this manuscript: Barbara Webb

**Have the authors made all data and (if applicable) computational code underlying the findings in their manuscript fully available?**

Reviewer #1: Yes

Reviewer #3: Yes

PLOS authors have the option to publish the peer review history of their article (what does this mean?). If published, this will include your full peer review and any attached files.

Reviewer #1: No

Reviewer #3: No

**Figure resubmission:**
---

## [Decision Letter · Decision Letter 1]

28 Oct 2025

PCOMPBIOL-D-25-00009R1

A data-driven biology-based network model reproduces *C. elegans*  premotor neural dynamics

PLOS Computational Biology

Dear Dr. Morrison,

Thank you for submitting your manuscript to PLOS Computational Biology. After careful consideration, we feel that it has merit but does not fully meet PLOS Computational Biology's publication criteria as it currently stands. Therefore, we invite you to submit a revised version of the manuscript that addresses the points raised during the review process.

Please submit your revised manuscript within 30 days Dec 28 2025 11:59PM. If you will need more time than this to complete your revisions, please reply to this message or contact the journal office at ploscompbiol@plos.org. Please include the following items when submitting your revised manuscript:

We look forward to receiving your revised manuscript.

Kind regards,

Barbara Webb

Academic Editor

PLOS Computational Biology

Hugues Berry

Section Editor

PLOS Computational Biology

**Additional Editor Comments:**

I agree with the review below that the additional analyses and revised discussion have substantially clarified the contribution of the paper. But I also agree that a more extended discussion of previous modelling in the introduction, to better set the context for this work, would be more appropriate.

**Reviewers' comments:**

Reviewer's Responses to Questions

**Comments to the Authors:**

Reviewer #1: The authors conducted additional analyses and clarified key features of the model. I am generally satisfied with the manuscript revisions addressing my comments, but I would like to add several minor comments.

1. Regarding the section “Promoter and suppressor signal neurons” added by the authors, the forward-reverse switching in their model appears to be highly dependent on input signals. The authors' response includes the following description:

" In the absence of any stimulus the neurons stabilize at different values with the forward neurons more active than the reversal neurons. Perturbing each neuron induces changes to the state of downstream neurons, as shown in Figure 7(d), emulating the experimental results reproduced in Figure 7(c). "

Including such descriptions not only in the response but also within the main text would likely facilitate readers' understanding of this model. It would also be beneficial to discuss whether experimental evidence supports these findings. Furthermore, if possible, I would like to see a description of what insights this model provides regarding the switching mechanism in real worms. Also, Toyoshima 2024 (doi: 10.1371/journal.pcbi.1011848) constructed a model focusing on neural dynamics based on whole-brain imaging data and structural connectivity, similar to this paper. However, in that model, switching of command interneuron activity occurs solely due to neural activity noise, without requiring any specific stimulus input. If possible, it might be worthwhile to discuss why this difference arises.

2. The response to the issue raised by Reviewer 3 in comment 4, in which the introduction does not address the fundamental differences from past similar mathematical model papers, appears insufficient. The revisions now include two sentences: "There is also a substantial theoretical literature, much of it consisting of phenomenological studies (see Discussion). With all of these available tools, we believe it is time to attempt a more systematic understanding of the neural mechanisms that drive function." However, the term ”phenomenological" alone does not clarify the specific strength of this paper compared to these previous studies, making the explanation insufficient. These differences should be properly explained in the introduction, not merely touched upon in the discussion.

3. The followings seem typos.

Around Line 406, “Signal input from dataset 2023-01-23-15 produces high frequency switches, resulting in shorter forward runs and reversals (Fig. 9(a)).” The dataset number appears to be incorrect.

Around Line 569 "identifiably" -> "identifiability"

**Have the authors made all data and (if applicable) computational code underlying the findings in their manuscript fully available?**

Reviewer #1: **No: **The code appears to be in its old state prior to revision.

PLOS authors have the option to publish the peer review history of their article (what does this mean?). If published, this will include your full peer review and any attached files.

Reviewer #1: No

**Figure resubmission:**
---

## [Editor Report · Decision Letter 2]

5 Dec 2025

Dear Morrison,

We are pleased to inform you that your manuscript 'A data-driven biology-based network model reproduces *C. elegans*  premotor neural dynamics' has been provisionally accepted for publication in PLOS Computational Biology.

Best regards,

Barbara Webb

Academic Editor

PLOS Computational Biology

Hugues Berry

Section Editor

PLOS Computational Biology

---

## [Editor Report · Acceptance letter]

PCOMPBIOL-D-25-00009R2

A data-driven biology-based network model reproduces *C. elegans*  premotor neural dynamics

Dear Dr Morrison,

I am pleased to inform you that your manuscript has been formally accepted for publication in PLOS Computational Biology. Your manuscript is now with our production department and you will be notified of the publication date in due course.

With kind regards,

Anita Estes
